# Integrins mediate symbiont-specific uptake in cnidarian larvae

Victor A S Jones[1,2,7], Melanie Dörr [ID][1,3,7], Isabelle Siemers [ID][1,4], Sebastian Rupp [ID][1,5], Sami El Hilali [ID][5], Sara Brites[5], Joachim M Surm[5], Ira Maegele [ID][1,6], Sebastian G Gornik[1], Meghan Ferguson [ID][5✉] & Annika Guse [ID][1,5✉]

## Abstract

Endosymbiosis between dinoflagellate algae and cnidaria is fundamental for coral reef health. Appropriate symbiont selection is required for sufficient host nutrient acquisition and could be tailored to increase cnidarian stress tolerance. Previous research suggested glycan–lectin interactions facilitate symbiont uptake; however, blockage of such interactions does not fully inhibit symbiosis establishment, suggesting other receptors are at play. Here, we use a combination of cnidarian model systems and human cell lines to determine if phagocytic integrins facilitate symbiont recognition and uptake. Integrins are highly expressed in the gastrodermal tissue of the host, where symbiosis takes place, and symbiont uptake alters the expression of integrins and downstream signaling molecules. Blockage of integrin binding sites with competitor peptides reduces symbiont uptake, while uptake of non-symbiotic algae, or uptake in a non-symbiotic cnidarian, is unaffected. Finally, overexpression of phagocytic integrins in human cells increases symbiont uptake, and mutation of the active binding site abolishes uptake. Our findings reveal integrins as important receptors for symbiosis establishment and shed light on the evolutionary functions of integrins during phagocytosis.

**Keywords** Aiptasia; Cnidarian; Integrin; Phagocytosis; Symbiosis
**Subject Categories** Cell Adhesion, Polarity & Cytoskeleton; Evolution & Ecology

## Introduction

To increase the chances of survival in competitive oligotrophic environments, some organisms have evolved ways to work in cooperation to mutually thrive. In shallow nutrient-poor waters across the tropics, reef-building corals establish a mutualistic symbiosis with dinoflagellate algae of the family Symbiodiniaceae (LaJeunesse et al, 2018; Muscatine and Porter, 1977). Using bidirectional nutrient transfer, these endosymbiotic dinoflagellates, located inside host gastrodermal cells, can provide over 90% of the host's nutritional needs in the form of photosynthetically fixed carbon, while the algae receive inorganic nutrients and shelter from herbivores (Muscatine, 1990; Muscatine et al, 1984; Wernegreen, 2012; Yellowlees et al, 2008). The success of coral reef ecosystems is strictly reliant on this fundamental association and is greatly threatened by the anthropogenic climate crisis, causing increased rates of symbiosis breakdown and subsequent coral bleaching (Hoegh-Guldberg et al, 2007, 1987; Hughes et al, 2017).

If not challenged by extensive stress, host-dinoflagellate associations are stable over the lifetime of an individual host; however, most coral species produce aposymbiotic (symbiont-free) larvae that take up algae from their surroundings (Smith and Douglas, 1987). In this way, endosymbiosis must be re-established each generation, allowing corals to acquire symbionts specifically adapted to the local environments in which they settle (Baird et al, 2009; Davies et al, 2017; Yamashita et al, 2014). However, considering the broad and diverse abundance of microorganisms present during symbiosis onset, decisive mechanisms must exist to ensure symbiosis is established with the desired partner.

A series of complex selection criteria, termed "winnowing", are required to establish a stable symbiotic association with a suitable partner (Nyholm and McFall-Ngai, 2004). In the case of endosymbiosis, these steps span symbiont recognition, phagocytosis, post-phagocytosis selection, long-term immune evasion, and establishment of a stable niche within host cells (Davy et al, 2012). Genomic and cellular studies provide growing evidence that the first steps of winnowing likely resemble microbial invasion of animal and plant hosts (Davy et al, 2012; Nyholm and McFall-Ngai, 2004). Here, pattern recognition receptors (PRRs) on the host plasma membrane detect microbe-associated molecular patterns (MAMPs) on the surface of microbial targets to initiate phagocytosis of symbionts in a receptor-mediated and specific manner (Uribe-Querol and Rosales, 2020). However, it has yet to be determined which MAMP-PRR interactions would be involved in symbiont-specific phagocytosis since multiple molecular mechanisms have been proposed.

[1]Centre for Organismal Studies (COS), Heidelberg University, Heidelberg 69117, Germany. [2]Prolific Machines Inc., 6400 Hollis St, Emeryville, CA 94608, USA. [3]Department of Biology, University of Konstanz, Konstanz 78464, Germany. [4]Department of Zoology, Stockholm University, Stockholm 10691, Sweden. [5]Quantitative Organismic Networks, Ludwig-Maximilian-Universität (LMU) Biocenter, Planegg-Martinsried 82152, Germany. [6]Michael Sars Centre, University of Bergen, Bergen 5006, Norway. [7]These authors contributed equally: Victor A S Jones, Melanie Dörr. ✉E-mail: ferguson@bio.lmu.de; annika.guse@biologie.uni-muenchen.de

Glycan–lectin interactions are among the best-studied pairings involved in inter-partner recognition in Cnidaria. They are common MAMP-PRR interactions in innate immune responses and mutualistic endosymbiosis, and have been extensively explored in the context of coral–algal symbiosis (Kilpatrick, 2002; McGuinness et al, 2003; van Rhijn et al, 2001). Various glycan-binding lectins are present in corals and detect glycans on the symbiont surface (Markell et al, 1992; Tortorelli et al, 2021; Wood-Charlson et al, 2006; Jimbo et al, 2000; Kuniya et al, 2015; Kvennefors et al, 2008; Takeuchi et al, 2017). Surface glycan profiles differ among symbiont clades and strains, potentially providing a basis for symbiont-specific selection (Logan et al, 2010; Markell and Wood-Charlson, 2010; Tortorelli et al, 2021). Research on the role of glycan–lectin interactions in coral–algal symbiosis is mostly based on enzymatic cleavage of glycan residues on symbionts, lectin addition to mask symbiont glycans, or lectin antibodies and glycans to competitively block host lectin-binding sites (Kuniya et al, 2015; Lin et al, 2000; Takeuchi et al, 2021, 2017; Wood-Charlson et al, 2006). However, broad digestion of cell-surface proteins significantly decreased symbiont uptake in *Fungia scutaria* larvae, while digestion with glycan-specific N-glycosidase had little effect on symbiont uptake (Wood-Charlson et al, 2006). In addition, symbiont uptake is not always impaired when symbiont surface glycans are masked with exogenous lectins, and surface glycan profiles were shown to differ only subtly between compatible and incompatible symbiont strains, providing no foundation to explain species-specific host colonization rates (Parkinson et al, 2018). Considering that other MAMP-PRRs have also been described to contribute to symbiont phagocytosis, such as complement and scavenger receptors (Lin et al, 2000; Wood-Charlson et al, 2006; Kvennefors et al, 2010; Neubauer et al, 2016; Poole et al, 2016), it appears that no singular recognition process governs the first step of winnowing and an interplay between various recognition pathways seems plausible. In addition to the hosts ability to distinguish symbionts using an assortment of PRRs, selection also occurs post phagocytosis. For instance, symbionts persist in the host by inhibiting immune processes that would otherwise lead to their expulsion, while non-symbiotic algae are incapable of inhibiting immunity and become expelled (Jacobovitz et al, 2021). Therefore, both specific and general uptake mechanisms likely exist in addition to multiple levels of post-phagocytic selection processes.

Promising candidates for additional receptors involved in symbiont recognition are integrins. Integrins are heterodimeric transmembrane cell-surface receptors that contain non-covalently associated alpha and beta subunits. Integrins most prominently function as receptors for cell-extracellular matrix (ECM) and cell-cell adhesion, but also have established roles in cell polarity, cell motility, and phagocytosis (Hynes, 2002). Integrins and their extracellular ligands in mammals are clustered into four main classes: arginine-glycine-aspartic acid (RGD)-binding, laminin-binding, leucine-aspartic acid-valine (LDV)-binding, and collagen-binding integrins (Humphries et al, 2006). Some integrins, such as RGD-binding and laminin-binding integrins, are ancient and present throughout Metazoans (Hynes, 2002). Interestingly, RGD-binding integrins initiate the phagocytosis of microbes and apoptotic cells in *Drosophila*, *C. elegans*, and humans, making them compelling candidates for mediating symbiont phagocytosis in cnidarian–algal symbiosis (Dupuy and Caron, 2008; Torres-Gomez

et al, 2020; Mrakovcic et al, 2023). Integrin-mediated phagocytosis is also a method of immune evasion for some pathogens. For instance, *Bordetella pertussis* can bypass the typical $F_c$ receptor-mediated phagocytosis and destructive oxidative burst and instead survive in phagosomes through RGD integrin-mediated entry (Hellwig et al, 2001). Given the role of RGD-binding integrins in phagocytosis and the persistence of microbes inside a host cell, we sought to determine the function of integrins during symbiosis establishment in cnidarians.

In this study, we make use of a combination of experimental systems, including *Exaiptasia diaphana* (commonly Aiptasia), *Acropora digitifera*, *Nematostella vectensis*, as well as human cells, to address the fundamental question of how symbionts are recognized prior to uptake into cnidarian host cells. We identify integrins as novel receptor candidates and demonstrate that symbiont uptake, but not that of other algae, is enhanced by RGD-binding-integrin-mediated phagocytosis, where the binding of a symbiont-surface RGD-motif adds specificity to the efficient uptake of beneficial partners.

## Results and discussion

### Symbionts alter integrin expression in a cell-intrinsic manner

Several pathogens rely on receptor–ligand interactions to gain entry into cells, including several types of integrin interactions (Mrakovcic et al, 2023). To determine if integrins may be involved in symbiosis establishment in Aiptasia, we first analyzed the genome (GCF_001417965.1) and identified integrins based on protein domain analysis (Fig. 1A). Specifically, two proteins featured at least one integrin beta domain, and four proteins contained at least one integrin alpha domain. Both beta integrins, ITB1 and ITB2, had a complete set of functional domains, while only one of the four predicted alpha integrins had a complete functional set of domains (ITA2). The remaining three alpha integrin proteins were truncated in the refseq gene models, as none of them contained a signal peptide that localizes them to the membrane. These remaining alpha integrins were manually curated based on another available gene model (GCA_001417965.1) to include a functional set of protein domains, with signal peptides, and used for downstream analysis (Fig. 1A). Using these updated gene models, we reanalyzed publicly available RNA-sequencing data from whole larvae that were exposed to the Aiptasia symbiont SSB01 (*Breviolum minutum*) and compared their gene expression to aposymbiotic larvae never exposed to symbionts (Fig. 1B) (Wolfowicz et al, 2016; Baumgarten et al, 2015; Data ref: Baumgarten et al, 2015). All genes that encode integrin subunits were downregulated in symbiotic larvae; however, only *ITA2* and *ITA3* showed significant downregulation (Fig. 1C; Dataset EV1). In addition, we found that *vinculin*, which encodes an adapter protein that links integrins with the cytoskeleton, was significantly downregulated in symbiotic larvae. As previously reported, multiple genes that encode NPC2, which facilitate sterol transfer from the symbiont to the host, were significantly upregulated (Fig. 1C) (Lehnert et al, 2014; Hambleton et al, 2019; Wolfowicz et al, 2016). This suggests that upon symbiosis establishment, integrins and their associated genes are

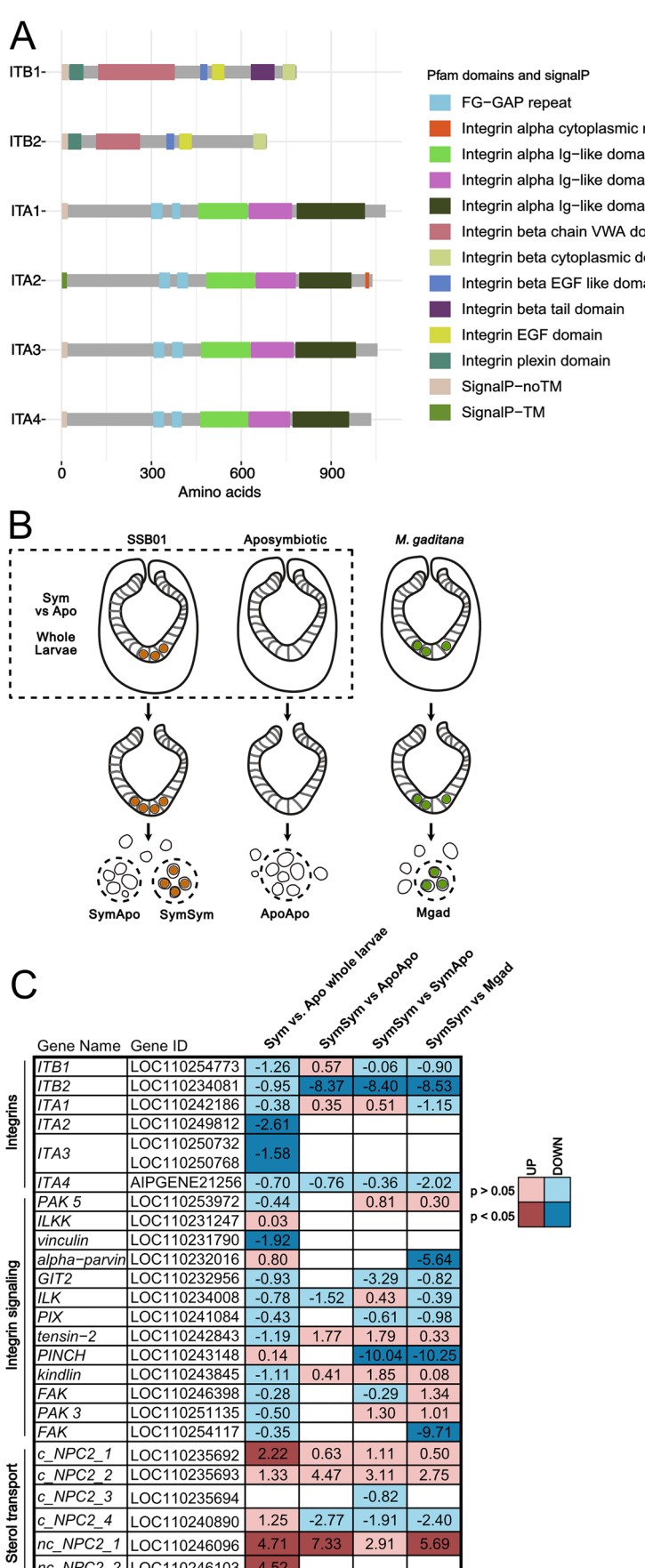

**Figure 1. Symbionts alter integrin expression in a cell-intrinsic manner.**

(A) Aiptasia integrin proteins and their respective domains. (B) Experimental setup for RNA sequencing analysis. Whole Aiptasia larvae with or without SSB01 symbionts were processed for bulk RNA sequencing in (Wolfowicz et al, 2016) and reused here. Aiptasia larvae exposed to either SSB01, *M. gaditana*, or left aposymbiotic for 24 h were manually dissected to extract gastrodermal cells with or without intracellular algae. The resulting samples were processed for RNA sequencing in (Jacobovitz et al, 2021) and reused here. (C) Expression changes in integrins, their downstream signaling genes, and NPC sterol transporters are shown. The colors and numbers in the heatmap indicate the log$_2$ fold change according to DESeq2 (blue = downregulation and red = upregulation). Empty fields indicate genes that were not detected in at least two replicates. Significantly differentially expressed genes ($P < 0.05$) are shown in darker colors, while non-significant genes are lighter colors. Source data are available online for this figure.

downregulated, while, e.g., sterol transporters representing symbiosis-specific marker genes become upregulated.

To determine if integrin downregulation is a global effect or if it is specific to the cells that house symbionts, we utilized our previous dataset where we dissected larval gastrodermal cells that contained SSB01 symbionts (SymSym) and compared their gene expression to neighboring gastrodermal cells without symbionts (SymApo), or to gastrodermal cells from aposymbiotic larvae (ApoApo) (Fig. 1B) (Jacobovitz et al, 2021; Data ref: Jacobovitz et al, 2021). Gastrodermal cells that phagocytosed symbionts had significantly decreased expression of *ITB2* compared to either aposymbiotic control (SymSym vs. ApoApo and SymSym vs. SymApo) (Fig. 1C; Dataset EV1). Furthermore, comparison of cells containing symbionts to those containing the non-symbiotic microalgae *Microchloropsis gaditana* (Mgad), which are lost relatively rapidly after uptake yet remain intracellular long enough for RNA-sequencing (Jacobovitz et al, 2021), revealed that symbiont-containing cells had decreased expression of *ITB2* and multiple downstream integrin signaling genes such as *alpha parvin* and *PINCH*, which encode proteins that form a complex with integrin-linked-kinase (ILK) to transduce integrin signals (SymSym vs. Mgad) (Fig. 1C; Dataset EV1) (Green and Brown, 2019). These data suggest that SSB01 decreases the expression of integrins and downstream signaling molecules in a cell-intrinsic manner, whereas non-symbiotic algae are incapable of altering integrin expression. We hypothesize that integrin signaling is downregulated upon symbiont uptake in a cell-intrinsic manner to avoid uptake of multiple symbionts into individual host cells, while this would not necessarily inhibit uptake into neighboring cells. Alternatively, integrins not involved in phagocytosis may be downregulated, while those involved in phagocytosis maintain their expression to prioritize symbiosis establishment. Regardless of the hypothesis, these data suggest that integrins play a role in symbiont acquisition.

## Predicted RGD-binding *integrin alpha 1* is highly expressed in Aiptasia larval gastrodermal tissue

To gain insight into the functional specificity of cnidarian integrins, we compared the phylogeny of alpha integrins from Aiptasia and other cnidarians to that of vertebrate integrins with known ligands. We focused on alpha integrins since these subunits primarily determine ligand specificity (Mezu-Ndubuisi and Maheshwari, 2021). Phylogenomic analysis revealed four integrin alpha subunits in Aiptasia (Figs. 2A and EV1), with cnidarian ITAs falling into two cnidarian-specific clades. We propose the names cnidarian ITA Clade 1 and 2 for these. Cnidarian ITA Clade 1 contains ITA1 (NP_001421589.1), as well as a single protein from each cnidarian species included in the analysis (Fig. 2A).

Cnidarian ITA Clade 1 falls together with two other previously well-supported clades with representatives from vertebrates: PS1, which canonically bind to laminins; and PS2, which canonically bind to the tripeptide arginine-glycine-aspartic acid (RGD) and the related lysin-glycine-aspartic acid (KGD) peptide (Fig. 2A) (Dupuy and Caron, 2008; Horton, 1997). The three other ITAs from Aiptasia belong to cnidarian ITA Clade 2, which likewise contains several integrins from Cnidarian species. This clade falls between the PS2 clades and clades containing the vertebrate α4/α9 cluster, which have been reported to bind the tripeptide leucine-aspartic acid-valine (LDV) (Fig. 2A) (Humphries et al, 2006). Taken together, the phylogenetic analysis suggests the alpha subunit ITA1 to potentially bind RGD-containing proteins, while other Aiptasia ITAs may bind to LDV or unknown ligands.

Interestingly, multiple human pathogens, including *Staphylococcus aureus*, *Yersinia spp.*, and *Bordetella pertussis*, have co-opted RGD-integrin interactions to adhere to, enter, enhance colonization and replication, and spread within the host (Mrakovcic et al, 2023). Given the role of RGD-binding integrins in the phagocytosis and persistence of microbes inside a host cell, we hypothesized that symbionts may also be phagocytosed and maintained by the host using similar mechanisms. Therefore, we localized the transcripts of the predicted RGD-binding integrin ITA1 by chromogenic in situ hybridization (ISH) and fluorescence in situ hybridization (FISH). We found that *ITA1* is specifically expressed in the larval gastroderm (Fig. 2B–E). Since symbionts are phagocytosed by gastrodermal cells, this finding is consistent with a possible role for ITA1 in symbiont uptake.

## RGD-integrins facilitate symbiont uptake in symbiotic cnidaria

To assess the role of integrins in symbiont uptake, we exposed aposymbiotic Aiptasia larvae to peptides containing known integrin recognition sequences, LDV or RGD, and the reverse control peptide DGR, to competitively block ligand binding sites before and during SSB01 exposure (Figs. 3A,B and EV2A,B). Incubation with either LDV or DGR did not significantly affect the number of intracellular symbionts. In contrast, RGD-peptide treatment decreased the mean intracellular SSB01 count per larva in a concentration-dependent manner, reducing it to 58% of the uptake seen under DGR-peptide-treated or LDV-peptide-treated conditions (Figs. 3B and EV2A). Although trends among replicates suggest that RGD blockage slightly decreases the percent of larvae that take up SSB01, these changes are not significant (Fig. EV2B). Bay et al obtained similar results upon general cleavage of symbiont surface molecules or by cleavage/masking of host glycan receptors. They suggested that the reduction in symbionts per larva

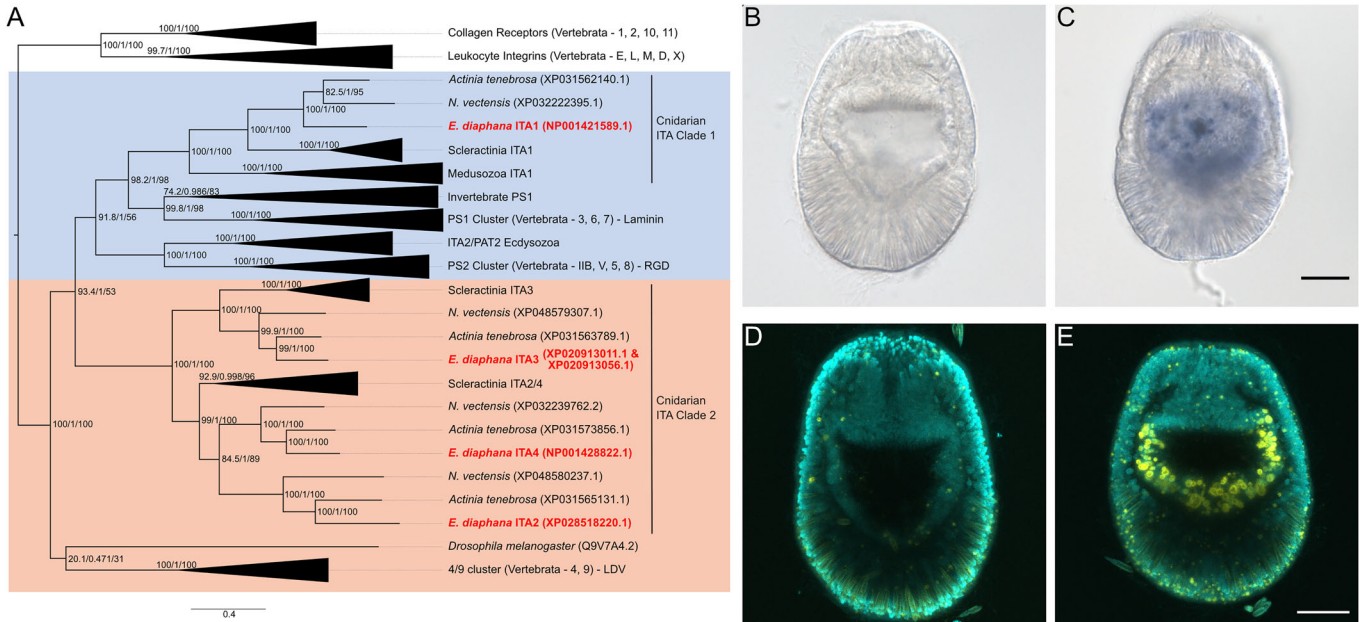

**Figure 2. Predicted RGD-binding *integrin alpha 1* is highly expressed in Aiptasia larval gastrodermal tissue.**

(A) Phylogeny based on protein sequences of cnidarian and non-cnidarian integrin alpha sequences. Aiptasia integrin alpha 1 (ITA1) clusters with known RGD and laminin-binding integrins (blue background). The other Aiptasia integrins (ITA2, ITA3, ITA4) cluster in a second group of cnidarian integrins (cnidarian ITA cluster 2) related to LDV-binding integrins (peach background). Numbers on branches and scale bar indicate substitutions per site. (B–D) Localization of *ITA1* expression in Aiptasia larvae. *ITA1* mRNA was detected with an appropriate antisense probe (C, E) but was absent with a sense probe (B, D) in in situ hybridization (B, C) and fluorescent in situ hybridization (D, E, yellow signal). Scale bars = 20 µm, cyan (C, E) indicates nuclei. Source data are available online for this figure.

was more due to decreased post-phagocytosis retention rather than pre-phagocytosis recognition (Bay et al, 2011). This is in line with the observed flexibility in the uptake of different Symbiodiniaceae species and even non-symbiotic algae, the latter of which eventually become expelled (Jacobovitz et al, 2021; Baird et al, 2007). Thus, it is possible that integrin-dependent signaling pathways, which are known to have a multitude of effects on cell motility, cytoskeletal organization, transcription control, proliferation, and cell survival, play an important role in both recognition and maintenance of the symbiont (Hynes, 2002). This is in line with the idea that signalling might be the more ancestral role of integrins rather than cell-adhesion, a function that is predominantly observed in higher animals (Sebé-Pedrós and Ruiz-Trillo, 2010).

To further investigate the role of RGD as a recognition motif, we coated inert polystyrene beads comparable in size to symbionts (~ 8 µm diameter) with either RGD- or DGR-peptides via an oligo-PEG linker, which were then incubated with Aiptasia larvae (Fig. 3C,D). After 24 h, RGD-coated beads were present inside larvae in higher amounts and in a higher proportion of larvae than beads coated with the control peptide (Figs. 3D and EV2C) and were seen phagocytosed by gastrodermal cells (Fig. EV2D). The enhanced uptake or increased early retention of particles coated with RGD-containing peptides indicates that an RGD-integrin interaction is sufficient for phagocytosis.

To explore whether RGD-integrin interactions also facilitate symbiont acquisition in corals, we assessed internalization of SSB01 in *Acropora digitifera* larvae while competitively blocking with either the RGD peptide or the control DGR peptide

(Fig. 3E,F). The number of SSB01 inside *A. digitifera* larvae significantly decreased after exposure to 1000 µM RGD-peptide when compared to the control DGR-peptide (Fig. 3F). Furthermore, we found that significantly more beads were inside coral larvae when coated with RGD compared to DGR (Fig. 3G,H). Taken together, these data suggest that RGD-binding integrins promote symbiosis and that this interaction is conserved between anemones and corals.

To characterize the evolutionary conservation of RGD-dependent enhancement of symbiont uptake, we investigated whether the mechanism is also present in the anemone *Nematostella vectensis*. While not naturally symbiotic, we observed that *N. vectensis* polyps (tentacle bud stage) take up low amounts of different microalgae (Fig. 3I,J). To examine the role of the RGD-integrin interaction in the uptake of SSB01 in *N. vectensis* we assayed competitive peptide blocking using a range of peptide concentrations (Figs. 3K and EV2E). Notably, the mean SSB01 numbers per polyp were not significantly different between RGD-peptide treatment and control conditions. These results suggest that the uptake of SSB01 is not dependent on recognition by RGD-binding integrins in the non-symbiotic sea anemone *N. vectensis*. Together, this indicates that RGD-integrin interactions are utilized by symbiotic Cnidaria to increase symbiont uptake specificity, whereas non-symbiotic Cnidaria rely on more general uptake mechanisms.

As previously mentioned, Aiptasia larvae can phagocytose various microalgae species effectively, both within Symbiodiniaceae and non-symbiotic microalgae (Wolfowicz et al, 2016; Jacobovitz et al, 2021). Hence, we sought to examine whether microalgae

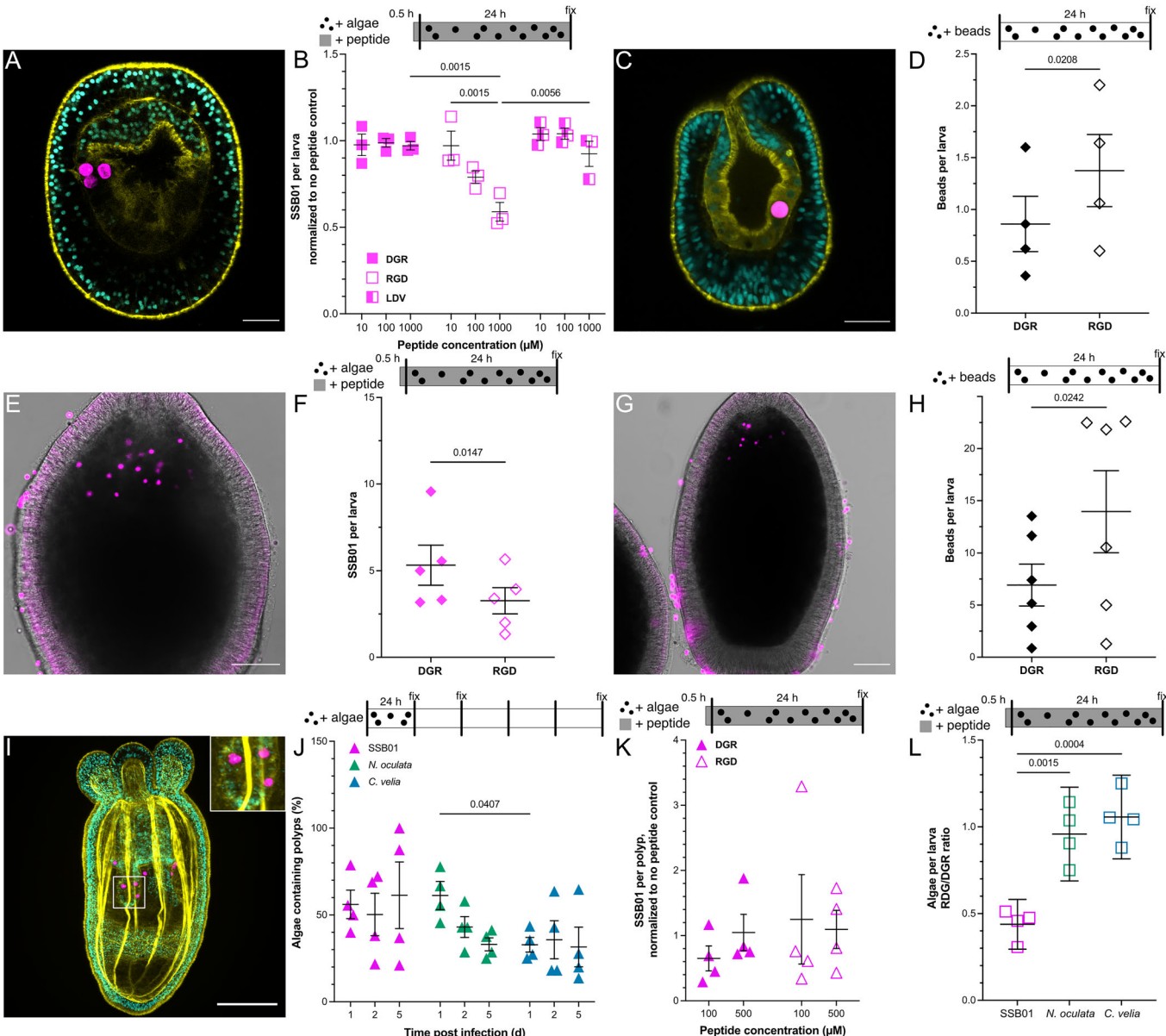

**Figure 3. RGD-integrins facilitate symbiont uptake in symbiotic cnidaria.**

(A) Aiptasia larvae with SSB01 symbionts (pink) treated with the control peptide DGR. Cyan = nuclei, yellow = actin. Scale bar = 20 μm. (B) SSB01 internalized by Aiptasia larvae normalized to a no-peptide control. Aiptasia larvae were exposed to control peptide DGR, integrin ligand RGD, or integrin ligand LDV for 30 min, then exposed to symbionts (*B. minutum*, SSB01) for 24 h. n = 3. (C) Aiptasia larvae with an internalized inert bead (pink) coated with RGD. Cyan = nuclei, yellow = actin. Scale bar = 20 μm. (D) Number of peptide-coated beads taken up by Aiptasia larvae. Beads were coated with either control peptide DGR or integrin ligand RGD, then incubated with Aiptasia larvae for 24 h. n = 4. (E) *Acropora digitifera* larvae with internalized SSB01 symbionts (pink). Scale bar = 100 μm. (F) SSB01 per *A. digitifera* larvae after a 24 h incubation and exposure to either the control peptide DGR or integrin ligand RGD (1000 μM peptide). n = 5. (G) *A. digitifera* larvae with internalized inert beads (pink) after 24 h of exposure. Scale bar = 100 μm. (H) Control peptide DGR or integrin ligand RGD-coated beads per *A. digitifera* larvae after 24 h exposure. n = 6 (I) Non-symbiotic *Nematostella vectensis* polyp with SSB01 (pink). Cyan = nuclei, yellow = actin. Scale bar = 100 μm. (J) Time course of microalgae uptake in *N. vectensis* polyps upon SSB01, *N. oculata* and *C. velia* exposure. n = 4. (K) SSB01 per *N. vectensis* polyp after a 24 h incubation and exposure to either the control peptide DGR or integrin ligand RGD. n = 4. (L) Algae per Aiptasia larvae as a ratio of RGD- to DGR-treated (1000 μM peptide). Less than 1 indicates RGD decreases algae per larvae compared to the DGR control. n = 4. For all plots, whiskers depict mean ± SEM. For (B, J–L), significance was found via ANOVA. For (D, F, H), statistical significance was found via paired *t* test. The number of biological replicates is stated for each quantification. Source data are available online for this figure.

uptake through integrin-dependent phagocytosis was symbiont-specific. To test whether inhibition of integrin binding broadly affects microalgae uptake, we repeated peptide blocking experiments in Aiptasia larvae using SSB01 symbionts and the non-symbiotic microalgae *Nannochloropsis oculata* and *Chromera velia* (Figs. 3L and EV2F–H). Here, SSB01 represents a true Aiptasia symbiont, *N. oculata* represents a distantly related algae, while *C. velia* is closely related to Symbiodiniaceae and has been

found to be associated, but not necessarily symbiotic, with corals (Mohamed et al, 2018; Cumbo et al, 2013). We found that SSB01 uptake was reduced after RGD exposure compared to the control DGR-peptide (Figs. 3L and EV2H). In contrast, we did not observe a significant reduction in the uptake of *N. oculata* or *C. velia* when larvae were treated with the inhibitory RGD peptide. These results indicate that integrin-dependent phagocytosis is symbiont-specific. Interestingly, blocking with RGD peptides only reduced SSB01 uptake/retention by ~50%. Likewise, RGD-coating of beads enhances uptake by ~50%, but a baseline uptake for other types of particles remains (Fig. 3D). This is consistent with a complex uptake mechanism involving multiple receptors, including glycan–lectin, as well as scavenging receptor interactions and a post-phagocytotic sorting mechanism during early symbiosis establishment (Wood-Charlson et al, 2006; Neubauer et al, 2016; Jacobovitz et al, 2021; Bay et al, 2011). However, the sequence of events and exact molecules involved remain to be elucidated.

## Integrin overexpression increases SSB01 uptake in HEK293T cells

To further establish whether RGD-binding integrins are involved in symbiont phagocytosis, we quantified SSB01 uptake in human embryonic kidney (HEK) 293T cells that overexpressed the mammalian RGD-binding integrin dimer αVβ3 (Dupuy and Caron, 2008; Horton, 1997; Torres-Gomez et al, 2020). To assess localization and co-expression of the two subunits, each subunit was tagged with one half of a split-YFP, which led to a robust signal at cell membranes, indicative of correct localization and complex formation (Fig. 4A). We found that overexpression of the mammalian RGD-binding integrins significantly increased the proportion of HEK cells that phagocytosed SSB01 compared to GFP-CaaX-expressing control cells, which allows for localization of GFP to the membrane (Fig. 4B,D). Notably, some intracellular SSB01 displayed a halo of YFP fluorescence, which indicates the integrin dimers decorated the phagolysosome membrane (Fig. 4A, inset). Taken together, SSB01 uptake in HEK cells is significantly enhanced by overexpression of RGD-binding integrins.

Integrins bind ligands in a pocket created between the alpha and beta subunits. The conserved cation-binding sites, Metal Ion-dependent Adhesion Site (MIDAS), Adjacent to MIDAS (AMIDAS), and Ligand-associated Metal-binding Site (LIMBS) in the beta-subunit play an important role in binding efficiency (Fig. EV3) (Emsley et al, 2000; Shimaoka et al, 2003; Valdramidou et al, 2008; Vorup-Jensen et al, 2003). Mutations in aspartic acid residues D119 and D217 (numeration according to processed human integrin beta 3) have been shown to decrease collagen binding, with collagen being the prototypic RGD-containing protein (Valdramidou et al, 2008). Thus, we generated a double mutant version (D119A, D217A) of the mammalian integrin β3 and used it in the previously described assay (Fig. EV3). Overexpression of the integrin β3 mutant, together with integrin αV, showed no difference in the integrin expression pattern (Fig. 4C); however, it resulted in a significantly lower fraction of HEK cells that phagocytosed SSB01 compared to the wild-type integrin dimer (Fig. 4D). These data suggest that integrins recognize a potential RGD-containing protein on the surface of SSB01 to enhance its uptake, a mechanism which is even conserved in human integrins.

To investigate whether RGD-binding integrins specifically facilitate the uptake of Aiptasia symbionts, we compared the uptake of SSB01, *N. oculata*, and *C. velia* in HEK cells after integrin overexpression (Fig. 4D). We found that SSB01 were taken up significantly more frequently than both *N. oculata* and *C. velia* in cells where RGD-binding integrins were overexpressed. Expression of the mutated integrin dimer led to particle uptake similar to the range of uptake in the GFP-CaaX-expressing control cells for all algae (Fig. 4D). Although integrin overexpression did significantly increase the number of cells that phagocytosed *C. velia* when compared to controls, the proportion is still much lower than that of cells containing SSB01 upon integrin overexpression (Fig. 4D). To determine whether the integrin dimer αVβ3 preferentially allows for uptake in an RGD-dependent manner, we coated beads with either RGD or DGR and compared bead uptake in cells overexpressing αVβ3 integrins to GFP-CaaX-expressing cells. RGD-coated beads were taken up by integrin-overexpressing cells approximately ten times more efficiently than DGR-coated beads or by cells in which integrins were not overexpressed (Fig. EV4). This tenfold increase in uptake mirrors the approximate tenfold increase in SSB01 uptake upon integrin overexpression. Our observations indicate that (1) integrin overexpression enhances SSB01-specific uptake and that (2) mutations in RGD-binding sites negatively impact SSB01 uptake, which strongly implicates RGD-binding integrins in successful symbiont phagocytosis and symbiosis establishment.

Together, integrin-dependent uptake appears to add specificity, as overexpression of integrins significantly increased the phagocytosis of SSB01 symbionts but not non-symbiotic algae. In addition, blocking with RGD peptides only affects symbiont uptake, and does not play a role in the uptake of non-symbiotic algae or in a non-symbiotic host (*N. vectensis*). Similarly, blocking certain glycan/lectin interactions reduces symbiont colonization, but has little effect on incompatible symbionts (Tortorelli et al, 2021). Being able to distinguish between beneficial and potentially parasitic or ineffective partnerships is a critical step in establishing a symbiotic relationship. Our data suggest integrin–ligand interactions to be a potential mechanism to facilitate the uptake of beneficial symbionts. However, non-symbiotic algae are still phagocytosed and establish a transient relationship inside host cells until finally being expelled (Jacobovitz et al, 2021). Therefore, it is unclear whether integrins play a role at the cell surface in the detection of symbionts and facilitation of uptake, or if integrins within the phagocytic vesicle signal to the host whether to expel or incorporate the algae. It is possible that integrins play a role in both steps of symbiosis establishment, uptake and post-phagocytic processing, which is an area of potential future research. In addition, the exact integrin dimer and symbiont cell-surface ligand involved in the selection of symbionts is unknown. It is also possible that the host secretes components that bind to symbiont cell wall ligands that are in conjunction recognized by host integrins to facilitate uptake and add an additional layer of specificity into the selection process of beneficial partners, as similar processes have been seen with glycan/lectin interactions in *Xenia* soft corals (Hu et al, 2023). Further examination is required to identify integrin ligand pairs, which would be beneficial in the quest to generate optimal host–symbiont pairings.

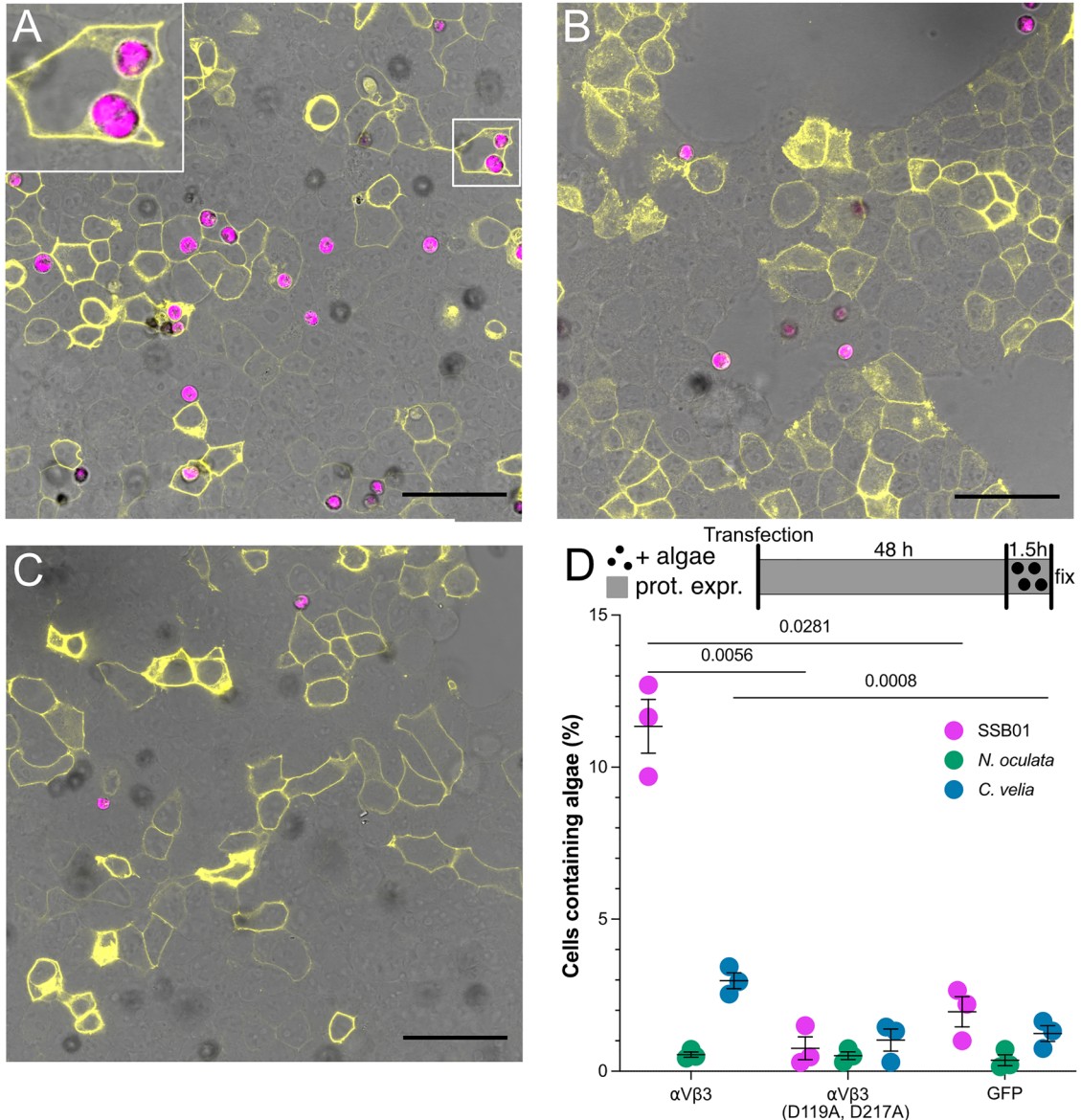

**Figure 4. Integrin overexpression increases SSB01 uptake in HEK293T cells.**

(A) HEK cells transfected with expression plasmids encoding mammalian integrin αV and β3 (each with halves of a split YFP and upon heterodimer formation fluoresce (yellow)) and exposed to SSB01 (pink). The inset shows integrins localized to the phagolysosome membrane. (B) HEK cells transfected with control plasmid encoding GFP-CaaX (yellow) exposed to SSB01 (pink). (C) HEK cells transfected with expression plasmids encoding mammalian αVβ3 integrins with the binding site mutated (D119A, D217A) (each with halves of a split YFP and upon heterodimer formation fluoresce (yellow)) and exposed to SSB01 (pink). Scale bars for (A–C) = 50 μm. (D) Percentage of HEK cells transfected with either αVβ3 integrins, αVβ3 integrins with the binding site mutated (D119A, D217A), or GFP-CaaX as a control that phagocytosed SSB01, *N. oculata*, and *C. velia*. For each condition, three biological replicates were used. Whiskers depict the mean ± SEM. Statistical significance was found via ANOVA followed by Tukey. For all images and plots protein expression was carried out for 48 h prior to a 90 min exposure to the respective algae. Source data are available online for this figure.

# Methods

### Reagents and tools table

| Reagent/resource | Reference or source | Identifier or catalog number |
| --- | --- | --- |
| **Experimental models** | | |
| *Breviolum minutum* (SSB01) | Xiang et al, 2013 | NA |

| Reagent/resource | Reference or source | Identifier or catalog number |
| --- | --- | --- |
| *Microchloropsis gaditana* | National Center for Marine Algae and Microbiota, Bigelow Laboratory for Ocean Sciences | CCMP526 |
| *Nannochloropsis oculata* | Norwegian Culture Collection of Algae K-1276; Norwegian Institute for Water Research | |

| Reagent/resource | Reference or source | Identifier or catalog number |
|---|---|---|
| *Chromera velia* | Norwegian Culture Collection of Algae K-1276; Norwegian Institute for Water Research | |
| *Exaiptasia diaphana* (F003 & CC7) | Carolina Biological Supply Company | 162865 |
| *Nematostella vectensis* | | |
| *Acropora digitifera* | | |
| HEK293T cells | American Type Culture Collection | CRL-3216 |
| **Recombinant DNA** | | |
| Plasmids for (F)ISH and HEK cell experiments | This study | Table EV1 |
| **Oligonucleotides and other sequence-based reagents** | | |
| Primers for cloning | This study | Table EV2 |
| **Chemicals, enzymes, and other reagents** | | |
| Daigo's IMK medium | Wako Pure Chemical Corporation | 398-01333 |
| PRO reef seasalt | Tropic Marin | 00005945 |
| Artemia cysts | Ocean nutrition | 4727 |
| Gibco™ Dulbecco's Modified Eagle's Medium | ThermoFisher Scientific | 41966-029 |
| Gibco™ Fetal Bovine Serum | ThermoFisher Scientific | 10500-064 |
| Gibco™ 10,000 U/mL penicillin-streptomycin | ThermoFisher Scientific | 15140-122 |
| Gibco™ Trypsin-EDTA | ThermoFisher Scientific | 25200-056 |
| COMPEL™ Magnetic beads with COOH modification (excitation 480 nm, emission 520 nm) | Bangs Laboratories | UMDG003 |
| COMPEL™ Magnetic beads with COOH modification (excitation 360 nm, emission 450 nm) | Bangs Laboratories | UMGB003 |
| MES buffer | Sigma-Aldrich | M8250 |
| ED(A)C (50 mg/mL, *N*-(3-Dimethylaminopropyl)-*N*'-ethylcarbodiimid-hydrochlorid) | Sigma-Aldrich | E6383 |
| Sulfo-NHS (50 mg/mL, sulfo-N-hydroxysuccinimide) | Abcam | Ab14569 |
| NH2-PEG8-Propionic acid | Sigma-Aldrich | JKA12005 |
| Tris (50 mM, pH 7.5) | Carl Roth | 4855.2 |
| SDGRG (H-Ser-Asp-Gly-Arg-Gly-OH) | Bachem | 4015321 |
| GRGDS (H-Gly-Arg-Gly-Asp-Ser-OH) | Bachem | 4008998 |
| EILV (FibronectinCS-1 Fragment | Bachem | 4026203 |
| Bovine serum albumin (BSA) | Sigma-Aldrich | A7906 |
| magnesium chloride hexahydrate ($MgCl_2$-6 $H_2O$) | Labochem | LC-5041.4 |
| Formaldehyde | Sigma-Aldrich | F1635 |

| Reagent/resource | Reference or source | Identifier or catalog number |
|---|---|---|
| Triton X-100 | Carl Roth | 3051 |
| Glycerol | Sigma-Aldrich | G5516 |
| 1,4-diazabicyclo[2.2.2] octane | DABCO; Sigma-Aldrich | D27802 |
| Non-toxic double-sided tape | Tesa | TES5338 |
| Tween-20 | Sigma-Aldrich | P7949 |
| DMSO | ThermoFisher Scientific | 67-68-5 |
| Phalloidin-Atto 565 | Sigma-Aldrich | 94072 |
| Hoescht 33258 | Sigma-Aldrich | B2883 |
| Sodium Azide | Sigma-Aldrich | S2002 |
| TRIzol | Ambion® by Life Technologies; ThermoFisher Scientific | 15596018 |
| Chloroform | Honeywell | 132950 |
| RNeasy Mini Kit columns | Qiagen | 74104 |
| SuperScript® IV Reverse Transcriptase | Invitrogen™; ThermoFisher Scientific™ | 18090010 |
| 5 U/µL *E. coli* RNase H | New England Biolabs Inc. | MO2975 |
| Q5® High-Fidelity DNA Polymerase | New England Biolabs Inc. | M0491 |
| GeneJET PCR Purification Kit | ThermoFisher Scientific | K0692 |
| T4 DNA ligase | New England Biolabs Inc. | M0202S |
| TOP10 chemically competent *E. coli* | ThermoFisher Scientific | C404010 |
| NEBuilder® HiFi DNA Assembly Cloning Kit | New England Biolabs Inc. | E5520S |
| DH5-alpha *E. coli* | ThermoFisher Scientific | 18265017 |
| Endura™ DUOs bacteria | Lucigen; Biosearch™ Technologies | 60240-0 |
| T4 DNA ligase buffer | New England Biolabs Inc. | B0202S |
| T4 Polynucleotide Kinase | New England Biolabs Inc. | M0201S |
| DpnI | New England Biolabs Inc. | R0176L |
| poly-L-lysin | Sigma-Aldrich | P8920 |
| AscI | New England Biolabs Inc. | R0558S |
| PacI | New England Biolabs Inc. | R0547S |
| Roche DIG RNA Labeling Kit | Roche | 11175025910 |
| phenol/chloroform/ isoamyl alcohol | Carl Roth | A156.1 |
| Sodium Acetate | Carl Roth | X891.1 |
| Isopropanol | Carl Roth | CP41.3 |
| Methanol | Carl Roth | X948.1 |
| Hydrogen peroxide | Merck | 1072090250 |
| Proteinase K | ThermoFisher Scientific | EO0492 |
| Glycine | Carl Roth | 0079.3 |
| Triethanolamine | Sigma-Aldrich | T9534-100G |
| Acetic anhydride | Sigma-Aldrich | 242845 |
| PFA, paraformaldehyde | Carl Roth | 0335.1 |
| Formamide | Carl Roth | P040.2 |
| Trisodium citrate | Carl Roth | HN12.2 |
| Sodium chloride | Thermo Fish Scientific | S/3161/60 |

| Reagent/resource | Reference or source | Identifier or catalog number |
|---|---|---|
| Heparin | Serva | 11761338 |
| Salmon sperm DNA | Invitrogen | 15632-011 |
| blocking buffer (Blocking solution, Roche in Maleic acid buffer) | Roche | 11096176001 |
| anti-Dig alkaline phosphatase | Roche | 11093274910 |
| anti-Dig horseradish peroxidase | Roche | 11207733910 |
| NBT/BCIP | Roche | 11681451001 |
| Tris | Carl Roth | 2449.2 |
| TSA Plus Fluorescein kit | Perkin Elmer | 16473824 |
| DABCO | Carl Roth | 0718.1 |
| **Software** | | |
| Leica LAS X | | Version 2.1.0 |
| Fiji | | Version1.53c |
| Excel | Microsoft | 16.16.6 |
| Geneious | | 10.2.6 |
| IQ-TREE | | 1.6.12 |
| FigTree | http://tree.bio.ed.ac.uk/software/figtree/ | |
| BBTools | https://sourceforge.net/projects/bbmap | 39.15 |
| R | | 4.4.3 |
| DESeq2 | https://doi.org/10.1186/s13059-014-0550-8 | |
| Prism | GraphPad Software, LLC | Version 9.1.1 |
| **Other** | | |
| Apogee PAR Quantum meter | Apogee | MQ-200 |
| Polycarbonate tanks | Cambro, Huntington Beach, CA, USA | GN 1/4 - 100 cm (# 44 CW) and 1/9 - 65 cm (#92 CW) |
| Intellus Ultra Controller Incubators | Percival | I-36LL4LX |
| Cell culture flasks | Cellstar®; Greiner Bio-One | 658195 |
| HERAcell™ 150i cell incubator | ThermoFisher Scientific | 50116047 |
| Leica TCS SP8 confocal microscope | Leica | |

## Methods and protocols

### Live organism and cell culture

Microalgae maintenance. *Breviolum minutum* clade B (family Symbiodiniaceae, strain SSB01; homologous Aiptasia symbiont) (Xiang et al, 2013), *Microchloropsis gaditana* CCMP526 (National Center for Marine Algae and Microbiota, Bigelow Laboratory for Ocean Sciences), *Nannochloropsis oculata*, and *Chromera velia* (Norwegian Culture Collection of Algae K-1276; Norwegian Institute for Water Research) were grown in cell culture flasks in 0.22-μm filter-sterilized Diago's IMK medium (Wako Pure Chemical Corporation). *B. minutum, N. oculata*, and *C. velia* were cultured at 26 °C, whereas *M. gaditana* was maintained at 18 °C. In all, 1–2 weeks prior to experiments, all microalgae cultures (including

*M. gaditana*) were split and kept at 26 °C on a 12 h light/12 h dark cycle under ~20–25 μmol m$^{-2}$ s$^{-1}$ of photosynthetically active radiation (PAR), as measured with an Apogee PAR Quantum meter (MQ-200; Apogee).

Aiptasia stock culture conditions. Clonal Aiptasia (*Exaiptasia diaphana*) lines F003 and CC7 (Carolina Biological Supply Company; 162865) were maintained in translucent polycarbonate tanks (GN 1/4 - 100 cm (# 44 CW) and 1/9 - 65 cm (#92 CW); Cambro, Huntington Beach, CA, USA) filled with artificial seawater (ASW; PRO-REEF Sea Salt, Tropic Marin®) at 31–34‰ salinity. Aiptasia stocks were kept in Intellus Ultra Controller Incubators (I-36LL4LX; Percival, Perry, USA) at 26 °C on a diurnal 12 h light:12 h dark cycle (12L:12D) under white fluorescent bulbs with an intensity of ~20–25 μmol m$^{-2}$ s$^{-1}$ of PAR and ~4000 K light temperature. Animals were fed twice per week using freshly hatched *Artemia* nauplii (Ocean Nutrition™) and cleaned 3 h later with cotton-tip swabs and tissue paper, followed by a water change.

Aiptasia spawning and larval culture conditions. Spawning of Aiptasia clonal lines F003 and CC7 was induced as described previously (Grawunder et al, 2015). Developing Aiptasia larvae were maintained in glass beakers in filter-sterilized artificial seawater (FASW) at 26 °C and exposed to a 12 L:12D cycle and ~4000 K light temperature.

*Nematostella* stock culture conditions. *Nematostella vectensis* stocks were cultured in polycarbonate tanks filled with 1/3 ASW at 11.0–11.5‰ salinity. Tanks were kept in darkness at 18 °C, and animals were fed once to twice per week with freshly hatched *Artemia* nauplii. *N. vectensis* were transferred into clean tanks filled with fresh 1/3 ASW (18 °C) once a week.

*N. vectensis* spawning and larval culture conditions. Spawning was induced as previously described (Fritzenwanker and Technau, 2002; Stefanik et al, 2013), with the following adaptations: Female and male tanks were rotated to spawn every 2–3 weeks. The day before spawning, animals were transferred to clean tanks with fresh 1/3 ASW (18 °C) and incubated for 8 h at 26 °C under white fluorescent bulbs with an intensity of ~20–25 μmol m$^{-2}$ s$^{-1}$ and ~4000 K light temperature. The water was replaced with fresh 1/3 ASW (18 °C) the next day and tanks were monitored for spawning for 2–3 h. Egg packages were transferred into petri dishes and fertilized with sperm water. Developing larvae were kept at 18 °C or 26 °C and filtered into fresh 1/3 FASW after escaping the jelly coat.

*Acropora digitifera* spawning. Colonies of the coral *Acropora digitifera* were collected off Sesoko Island (26°37'41"N, 127°51'38"E, Okinawa, Japan) according to Okinawa Prefecture permits and CITES export and import permits. They were handled as previously described (Wolfowicz et al, 2016) at the Sesoko Tropical Biosphere Research Center (University of Ryukyus, Okinawa, Japan). Isolated *Acropora* colonies were kept until spawning, and spawned symbiont-free gametes were mixed for fertilization. Planula larvae were then maintained at around 1000 larvae/L in filtered natural seawater, which was exchanged daily.

HEK293T cell culture conditions. Adherent HEK293T cells (American Type Culture Collection, VA, USA) were cultured in Gibco™

Dulbecco's Modified Eagle's Medium (DMEM; 41966-029; Thermo-Fisher Scientific™) supplemented with 10% (vol/vol) heat-inactivated Gibco™ Fetal Bovine Serum (FBS; 10500-064; Thermo-Fisher Scientific™) and 1% (vol/vol) Gibco™ 10,000 U/mL penicillin-streptomycin (15140-122; ThermoFisher Scientific™). Cells were grown in cell culture flasks (658195, Cellstar®; Greiner Bio-One) at 37 °C with 5% $CO_2$ in a HERAcell™ 150i cell incubator (50116047; ThermoFisher Scientific™) and passaged two to three times a week using 0.25% Gibco™ Trypsin-EDTA (25200-056; ThermoFisher Scientific™).

### Uptake assays

Bead coating. COMPEL™ Magnetic beads with COOH modification (Bangs Laboratories, UMDG003 (Fig. 3D,F,H), UMGB003 (Fig. EV4)) were coated with peptides and used for infections. Around 11 million Beads were washed three times in 400 µL MES buffer (0.05 M, pH 5, Sigma-Aldrich, M8250) and resuspended in 320 µL MES buffer. Next, 40 µL each of ED(A)C (50 mg/mL, N-(3-dimethylaminopropyl)-N'-ethylcarbodiimid-hydrochlorid, Sigma-Aldrich, E6383) and Sulfo-NHS (50 mg/mL, sulfo-N-hydroxysucci-nimide, Abcam, ab14569) were added and incubated with rotation at room temperature (RT) for 15 min, after which they were washed three times in 400 µL ice-cold MES buffer and resuspended in 390 µL ice-cold phosphate-buffered saline (PBS, pH 7.2). A linker was then added to the beads through the incubation with 10 µL of 100 mM NH2-PEG8-Propionic acid (Sigma-Aldrich, JKA12005) for 2 h at RT with rotation. After this, the beads were washed and incubated in 400 µL Tris (50 mM, pH 7.5, Carl Roth, 4855.2) for 15 min, and then washed three times in 400 µL MES buffer. At this stage beads can be stored before coupling to peptides. In all, 0.5 million beads were used for coupling and resuspended in 160 µL MES buffer. 20 µL each of ED(A)C and Sulfo-NHS (50 mg/ml each) were added and incubated at RT with rotation for 15 min, followed by three washes in MES buffer. Beads were resuspended in 99 µL PBS (pH 7.2) and 1 µL 100 mM of peptide, SDGRG (H-Ser-Asp-Gly-Arg-Gly-OH, Bachem, 4015321), GRGDS (H-Gly-Arg-Gly-Asp-Ser-OH, Bachem, 4008998) or EILV (FibronectinCS-1 Fragment, Bachem, 4026203) was added and incubated with rotation at RT for 2 h. Beads were then incubated at RT with rotation for 15 min in Tris (50 mM, pH 7.5), before three final washes in MES buffer and resuspension in FASW.

Exposure of Aiptasia larvae, A. digitifera larvae and N. vectensis to beads/algae. At least three replicates of Aiptasia larvae (4–8 dpf), A. digitifera larvae (3–6 dpf) or N. vectensis early tentacle bud stages (4–6 dpf) were collected in 1.5-mL bovine serum albumin-coated (BSA; A7906; Sigma-Aldrich) tubes. Where indicated 500 µL SDGRG, GRGDS, or EILDV peptide (stocks: 100 mM) solutions were prepared in FASW (Aiptasia) or 1/3 FASW (N. vectensis) at indicated concentrations. 50 N. vectensis tentacle buds or 300–500 Aiptasia larvae in 500 µL 1/3 FASW or FASW, respectively, were transferred to each peptide solution using 1% BSA-coated pipette tips and incubated for 30 min at 26 °C, when treated with peptides, otherwise this step was omitted. Exposure to algae or beads were performed at a final concentration of $5 \times 10^4$ cells/mL (Aiptasia/A. digitifera) or $1 \times 10^5$ cells/mL (N. vectensis) and incubated at 26 °C with rotation (1 rpm) and exposed to a 12 L:12D cycle for 24 h. Prior to fixation, N. vectensis polyps were relaxed in 344 mM magnesium chloride (LC-5041.4; Labochem® International) in 1/3 FASW for 10 min and transferred into 1% BSA-coated tubes.

Staining and mounting. Samples were fixed in 4% formaldehyde solution (F1635; Sigma-Aldrich) for 30 min and then washed twice in 0.1% Triton X-100 in phosphate-buffered saline (PBS-Triton) (3051; Carl Roth). They were either stained for F-actin and DNA (see below) or directly washed stepwise into glycerol from 30 to 50% and finally mounted in 87% glycerol (G5516; Sigma-Aldrich) in PBS with the addition of 2.5 mg/mL 1,4-diazabicyclo[2.2.2]octane (DABCO; D27802; Sigma-Aldrich). A lash sword was used to position the polyps along their lateral axis on the microscopy slide. Non-toxic double-sided tape (TES5338; tesa®) was used as a spacer between the microscopy slide and the coverslip. For staining, fixed Aiptasia larvae or N. vectensis polyps were washed 3 times in 0.05% Tween-20 (P7949; Sigma-Aldrich) in PBS (PBS-T) for 5 min. For permeabilization, samples were rotated at 0.25 rpm in 1% PBS-Triton and 20% dimethyl sulfoxide (DMSO; 67-68-5; ThermoFisher Scientific™) for 1 h at RT. After 3 washes in PBS-T for 10 min, larvae or polyps were incubated in Phalloidin-Atto 565 (94072; Sigma-Aldrich) diluted 1:200 in PBS-T for 1 h at 0.25 rpm in the dark. Samples were washed 3 times in PBS-T for 5 min before incubation with 10 µg/mL Hoechst 33258 (B2883; Sigma-Aldrich) diluted in Tris-buffered saline (pH 7.4), 0.1% Triton X-100, 2% BSA, and 0.1% sodium azide (S2002; Sigma-Aldrich) for 20–30 min at 0.25 rpm and RT in the dark. Larvae or polyps were washed three times for 5 min with PBS-T and then mounted as described above.

Microscopic analysis. Confocal microscopic analysis was carried out on a Leica TCS SP8 confocal laser scanning microscope using a ×10 dry immersion objective (numerical aperture = 0.30) or a ×63 glycerol immersion objective (numerical aperture = 1.30), Leica LAS X and Fiji software (version 2.1.0/1.53c) (Schindelin et al, 2012). Hoechst, Atto-565, and microalgae autofluorescence were excited with 405, 561, and 633 nm laser lines, respectively. Fluorescence emission was detected at 410–501 nm for Hoechst, 542–641 nm for Phalloidin-Atto 565, and 645–741 nm for symbiont autofluorescence.

Quantification of infection efficiency. The number of intracellular algae or particles was counted for at least 30 Aiptasia larvae, 20 A. digitifera larvae or 30 N. vectensis polyps per replicate per microalgal or particle type, and data recording was documented in Microsoft Excel version 16.16.6.

### Molecular cloning

RNA isolation/cDNA synthesis. In all, 1–2 CC7 polyps or 5000 aposymbiotic Aiptasia larvae (5–8 dpf) were homogenized until dissolved using 1 mL TRIzol® (15596018; ambion® by Life Technologies; ThermoFisher Scientific™) and a tissue homogenizer (Mini-Batch D-1; MICCRA). After samples were incubated at RT for 5 min, 200 µL Chloroform (132950; Honeywell) was added, and the samples were incubated for 3 min at RT before centrifugation (10,000–12,000 × g, 15 min, 4 °C). The supernatant was transferred into a fresh RNase-free 1.5-mL Eppendorf tube, and one volume of 70% ethanol was added. RNA was applied to RNeasy Mini Kit columns (Qiagen, 74104) and purified according to the manufacturer's instructions. cDNA was transcribed using the SuperScript® IV Reverse Transcriptase (Invitrogen™, 18090010; ThermoFisher Scientific™) following the manufacturer's instructions. Complementary RNA was removed by adding 0.5 µL of 5 U/µL E. coli RNase H (MO2975; New England Biolabs Inc. (NEB)), followed by incubation for 20 min at 37 °C.

**Restriction cloning.**   For cloning of plasmid P-0251, the insert was PCR-amplified from cDNA using Q5® High-Fidelity DNA Polymerase (M0491; NEB) with the primers and template defined in Table EV2. After column- or gel-purification using the GeneJET PCR Purification Kit (ThermoFisher Scientific™) according to manufacturer's manual, PCR product and vector (Tables EV1 and EV2) were digested using enzymes indicated in Table EV2, according to the manufacturer's instructions (New England Biolabs), ligated using T4 DNA ligase (NEB) according to the manufacturer's instructions and transformed to TOP10 chemically competent *E. coli* (ThermoFisher Scientific™) according to the manufacturer's instructions. The insert sequence was checked by sequencing.

**NEBuilder® HiFi DNA Assembly.**   For cloning of plasmids P-0299 and P-0300, fragments were amplified with primers from cDNA or plasmid templates as indicated in Table EV2, using Q5® High-Fidelity DNA Polymerase, per the manufacturer's instructions. After column- or gel-purification of inserts using the GeneJET PCR Purification Kit (ThermoFisher Scientific™) according to the manufacturer's manual, cloning vectors (Table EV1) were digested using enzymes indicated in Table EV2, according to the manufacturer's instructions (NEB). Template plasmids were removed by gel extraction using the GeneJET PCR Purification Kit. Vector and insert concentrations were determined by 1% agarose gel electrophoresis, and constructs were assembled using the NEBuilder® HiFi DNA Assembly Cloning Kit (E5520S; NEB) using a 1:2 vector:insert dsDNA pmols ratio following the manufacturer's instructions. Samples were diluted 1:4–1:3 prior to transformation of 2.5 μL into chemically competent DH5-alpha *E. coli* (18265017; ThermoFisher Scientific™) or Endura™ DUOs bacteria (60240-0; Lucigen; Biosearch™ Technologies). The insert sequence was checked by sequencing.

**Site-directed mutagenesis.**   For cloning of plasmids P-302 and P-0303, primers with mismatches (underlined in Table EV2) were used to amplify the original plasmid (Table EV1) using the Q5® High-Fidelity DNA Polymerase, per the manufacturer's instructions. Similar to site-directed mutagenesis kits, 1 μL of PCR product was used in a reaction mix containing, 1 μL of T4 DNA ligase buffer (B0202S; NEB), 1 μL T4 Polynucleotide Kinase (M0201S; NEB), 1 μL T4 ligase (M0202S; NEB), 1 μL DpnI (R0176L; NEB), and 5 μL $H_2O$, and incubated for 1 h at 37 °C. Half the reaction mix was transformed into chemically competent DH5-alpha *E. coli*.

### HEK293T cell infection assays

**Mammalian αVβ3 plasmid constructs.**   If not stated otherwise, 155 fmol human integrin subunit αV fused to the C-terminal part of a split-YFP (P-0299), and 167 fmol mouse integrin subunit β3 (P-0300) fused to the N-terminal part of a split-YFP were used to overexpress the integrin dimer αVβ3 in human HEK293T cells. In total, 80 fmol GFP-CaaX (pEGFP-f, AGP57; Clontech; obtained from Ary Shalizi, Stanford University, P-0273) together with (empty) 282 fmol pCS2+ plasmid constructs (P-0301, gift from Sergio Acebron, Heidelberg University) were transfected as a negative control. As a non-functional control, the beta 3 subunit was mutated by changing aspartic acid D119 and D217 to alanine (P-0303, based on (Valdramidou et al, 2008)), and overexpressed (167 fmol) together with the alpha subunit as indicated above.

**Calcium phosphate transfection.**   HEK293T cells ($0.75 \times 10^5$ cells/well) in DMEM were grown in 12-well plates (665102; Cellstar®; Greiner Bio-One) on sterile poly-L-lysin (0.01%, P8920; Sigma-Aldrich) coated coverslips overnight at 37 °C with 5% $CO_2$. Appropriate amounts of plasmid DNA constructs (see above) and 7.5 μL 2.5 M $CaCl_2$, were added to water to a final volume of 75 μL. An equal volume of 2× HeBS buffer (pH 7.05) was added dropwise, and the mix was incubated for 10 min at RT. The transfection mix was added, and cells were incubated at 37 °C with 5% $CO_2$ for 5–7 h before gently washing twice with 1× dPBS and subsequently adding fresh DMEM. Cells were incubated for 48 h prior to infection.

**Uptake in HEK293T cells.**   HEK293T cells were infected using symbionts, *N. oculata*, or *C. velia* 48 h post-transfection. In all, 1–2-week-old algae cultures in IMK medium were pelleted for 5 min at $2000 \times g$ and resuspended in DMEM at a final concentration of $3 \times 10^5$ cells/mL (if not stated otherwise). Prior to exposure to algae or beads, transfected HEK293T cells were washed once with pre-warmed 1× dPBS, before adding 1 mL of algae/beads suspension in each well. After incubation for 1.5 h at 37 °C with 5% $CO_2$, cells were washed once with pre-warmed 1× dPBS. HEK293T cells were fixed using 4% FA in 1× dPBS at RT for 30 min, washed with dPBS, and mounted on microscopy slides in 100% glycerol. Algal/bead uptake was compared in a minimum of three independent biological replicates.

**Quantification of HEK293T cell infection efficiency.**   The total number of transfected HEK293T cells and transfected cells with intracellular algae was counted in Z-stacks (size 20–40 μm) of six randomly chosen fields of view per sample. Confocal microscopic analysis was carried out on a Leica TCS SP8 confocal laser scanning microscope using a ×63 glycerol immersion objective (numerical aperture = 1.30), Leica LAS X and Fiji software (version 2.1.0/1.53c) (Schindelin et al, 2012). eGFP, YFP, and algae autofluorescence were excited using 488, 514, and 633 nm laser lines, respectively. Fluorescence emission was detected at 493–570 nm for eGFP, 519–590 nm for YFP, and 645–741 nm for microalgae autofluorescence.

### (Fluorescent) in situ hybridization

**Probe generation.**   The plasmid for probe-generation (Table EV1) was designed as described in the molecular cloning section. Digoxygenin-labeled probes were generated after linearization with either AscI (R0558S: NEB) or PacI (R0547S: NEB); dependent on final orientation of the probe, using DIG RNA Labeling Kit (11175025910: Roche). In short, 1 μg linearized plasmid was incubated with transcription buffer, NTP labeling mixture, RNase inhibitor, T7 or SP6 polymerase for 2 h at 37 °C, before adding DNaseI for an additional incubation of 20 min. RNA was extracted using phenol/chloroform/isoamyl alcohol (25:24:1) (A156.1: Carl Roth), and precipitated using 3 M sodium acetate (X891.1: Carl Roth) and isopropanol (CP41.3: Carl Roth).

**In situ hybridization.**   For in situ hybridization, Aiptasia larvae (3–8 dpf) were fixed in 3.7% PFA in FASW for 1 h, washed 3 times in PBS-Triton with 1% BSA, followed by three washes in 100% methanol (X948.1: Carl Roth). Larvae were then incubated in 90% methanol and 3% hydrogen peroxide (1072090250: Merck) in water for 30 min at RT, followed by rehydration through 60% methanol and 40% of 0.1% Tween-20 in PBS (PBS-T), 30%

methanol and 70% PBS-T, and four washes in 100% PBS-T for 5 min each. Next, larvae were incubated in 0.01 mg/mL proteinase K (E00492: ThermoFisher) in PBS-T for 8 min prior to two 3 min washes in 2 mg/mL glycine in PBS-T. Larvae were washed with 1% Triethanolamine (T9534: Sigma-Aldrich) in PBS-T, then with 0.3 µL/mL acetic anhydride (242845: Sigma-Aldrich) in 1% triethanolamine in PBS-T for 10 s, then with 0.6 µL/mL acetic anhydride in 1% triethanolamine in PBS-T for 10 s and then twice in PBS-T, followed by a 30 min incubation in 4% PFA in PBS-T at RT and five washes in PBS-T.

Next, larvae were incubated for 10 min in hyb buffer (50% formamide (P040.2: Carl Roth), 0.075 M trisodium citrate (HN12.2: Carl Roth), 0.75 M sodium chloride (S/3161/60: Thermo-Fisher), 0.05 mg/mL heperin (11761338: Serva), 0.25% Tween-20, 1% SDS, 0.05 mg/mL salmon sperm DNA (15632-011: Invitrogen), pH 4.5) at RT, followed by 1–4 h incubation in fresh hyb buffer at 60 °C. Larvae were then incubated with denatured (10 min at 90 °C) DIG-labeled probes (1 ng/µL) at 60 °C for 36–72 h with rotation. For competition with unlabeled probes, a tenfold excess was used. After incubation, larvae were washed at 60 °C for 5 min in hyb buffer, and again for 15 min in hyb buffer, prior to wash into 2X SSC buffer (0.3 M sodium chloride, 0.03 M trisodium citrate, pH 7.0) in consecutive steps of 75% hyb buffer with 25% 2× SSC buffer, followed by 50% each buffer and 25% hyb buffer with 75% 2× SSC buffer and finally 100% 2× SSC buffer for 10 min each at 60 °C followed by two 20 min washes at 60 °C in 0.05× SSC (7.5 mM sodium chloride, 0.75 mM trisodium citrate, pH 7.0). Next, larvae were washed into PBS-T in three steps at RT from 75% 0.05× SSC with 25% PBT to 50% each, 25% 0.05× SSC with 75% PBS-T to 100% PBS-T. After this, larvae were incubated in blocking buffer (11096176001: Roche) for 30 min at RT, followed by incubation in anti-Dig alkaline phosphatase (for in situ hybridization) (11093274910: Roche) or anti-Dig horseradish peroxidase (for fluorescent in situ hybridization) (11681451001: Roche) diluted 1:5000 in blocking buffer at 4 °C overnight, followed by three short washes in PBS-T and seven washes of 20–30 min at RT.

For detection of in situ hybridization, larvae were washed twice in AP buffer (0.1 M sodium chloride, 0.1 M tris (pH 9.5), 0.1% Tween-20) for 5 min at RT and twice in AP buffer with 0.05 M magnesium chloride for 5 min at RT. Larvae were then incubated in NBT/BCIP staining solution (0.1 M Tris-HCl (pH 9.5), 0.1 M sodium chloride, 1:50 NBT/BCIP stock (11681451001: Roche)) at 37 °C until sufficient staining was detected, at which point staining was stopped with an equal volume of 100% ethanol. This was followed by two washes in 100% ethanol and three washes in PBS-Triton, followed by a final wash with PBS, before mounting in 90% glycerol, 10% Tris (0.1 M, pH 8.0).

For the detection of fluorescent in situ hybridization, the TSA Plus kit (16473824: Perkin Elmer) was used. Larvae were incubated for 10 min in FITC stock solution diluted 1:50 in amplification diluent at RT, after which they were washed twice in PBS-Triton for 5 min at RT. For counterstaining of the nuclei, 10 µg/mL Hoechst was used for 10 min at RT, followed by three 10 min washes at RT in PBS-Triton and one wash in PBS. The samples were then mounted in 90% glycerol, 10% Tris (0.1 M, pH 8.0) with 2.6 mg/mL DABCO (0718.1: Carl Roth), and microscopically analyzed using a Nikon Eclipse microscope with a color camera.

## Integrin protein and transcriptomic analysis

Review integrin gene models. To identify the set of Aiptasia integrins, all predicted proteins from refseq were queried against the Pfam database (Paysan-Lafosse et al, 2025) using the hmmsearch function from HMMER-3.3 (Eddy, 2011). Two proteins featuring at least one "integrin_beta" domain and four proteins featuring at least one "integrin_alpha" domain were kept as candidates for careful inspection. A set of revised gene models and predicted proteins (see additional files in source data for Fig. 1A,C) was generated for the purpose of the RNA-seq analysis conducted in this study. The gene models of the two beta subunits were kept as is. The gene LOC110254773 was renamed *ITB1* as it was found to be the reciprocal best blast hit to the human integrin beta 1. The second beta-subunit LOC110234081 was named *ITB2* in the figures and annotation files. One alpha subunit was kept as it appears in the refseq annotation as it seemed complete in terms of the functional domains contained in the protein that it encodes (XP_028518220.1). This gene was renamed *ITA2*. The three other alpha subunit genes were truncated in the refseq gene models, as none of their predicted protein sequences contained a signal peptide. One curated model was generated in refseq for *ITA1*, and its protein sequence has the accession number NP_001421589.1. The initial gene model of *ITA4* submitted to genbank encoded for a protein featuring a signal peptide. This gene model (AIPGENE21256 which encodes the protein KXJ25122.1) was used to create a curated gene model in refseq for *ITA4* (accession number of the protein: NP_001428822.1). The alpha subunit *ITA3*, was split into two genes in the refseq gene models (LOC110250732, LOC110250768) with the first being very short and encoding a protein that features an N-terminal signal peptide (XP_020913011.1) while the second encodes for a protein featuring all the other domains but no signal peptide (XP_020913056.1). The initial gene model submitted by KAUST spans both genes but seems too long as it engulfs a third gene and its resulting predicted protein does not have a significant signal peptide anymore. We continued with a gene model manually created for *ITA3*. It spans the two genes mentioned above. One remaining gene was annotated in refseq as a partial integrin. It is located on a very small contig and by blasting it to the three higher quality genomes of CC7, RED, and HAW clonal lines accessible through the reefgenomics webportal (www.reefgenomics.org), we found that it points at the same genes as querying ITA3 against them. This suggests that this is possibly an assembly error in the initial reference genome, and we ignored this additional gene for this study. Pfam domains (Paysan-Lafosse et al, 2025) and significant signal peptides (Teufel et al, 2022) of the revised integrin subunits were then predicted using Interproscan online (Jones et al, 2014).

Integrin alpha (ITA) phylogeny. Identification of integrin α proteins was performed in Geneious 10.2.6 (https://www.geneious.com). Aiptasia integrin α proteins with refseq accession numbers NP_001421589.1, XP_028518220.1, XP_020913056.1 + XP_020913011.1 (combined into one gene for analysis), XP_020908361.1 and (we propose referring to these as ITA1, ITA2, ITA3, and ITA4 respectively) were used as starting point for a blastp search (in Geneious) against the refseq protein database, retrieving the 40 top hits for each, combining these to a total of 83 unique hits. Several of these were different isoforms which were manually curated to give a total of 49 unique hits. The same proteins together with the human ITAE protein (Uniprot accession P38570) were used for a blastp search against the uniprot database retrieving 30 top hits each, combined to a total of 55 unique hits. Sequences were aligned using the MUSCLE Alignment function with default parameters (in Geneious). The protein alignment was then imported into IQ-TREE (1.6.12) and

used to perform phylogenetic analysis (Nguyen et al, 2015). First, ModelFinder determined that the best mode of protein evolution was WAG + I + G4 model (Kalyaanamoorthy et al, 2017), using the Bayesian information criterion. Phylogenetic trees were then generated using 1000 ultrafast bootstrap iterations (Minh et al, 2013) and the SH-aLRT test (Anisimova et al, 2011). The tree was visualized using FigTree (http://tree.bio.ed.ac.uk/software/figtree/). The full tree (Fig. EV1) was then collapsed for ease of analysis.

RNA sequencing and analysis. Raw FastQ files from previously published RNA-seq samples of Aposymbiotic and Symbiotic Aiptasia larvae (Wolfowicz et al, 2016; Baumgarten et al, 2015; Data ref: Baumgarten et al, 2015) were downloaded from NCBI (accessions: SRR1648373-76). The FastQ files were mapped simultaneously to the Aiptasia reference genome (GCF_001417965.1) and the genome of Symbiodinium minutum (Shoguchi et al, 2013) using the script BBsplit from BBTools 39.15 https://sourceforge.net/projects/bbmap. Only reads that mapped exclusively to the Aiptasia genome were kept for further analysis. Filtered reads were then aligned to the Aiptasia genome using STAR (Dobin et al, 2013) with the option --quantMode set to GeneCounts. The gene models used to compute the GeneCounts were carefully curated as Integrin alpha genes needed to be manually re-annotated (see below). The differential gene expression analysis was then conducted in R (4.4.3) using the DESeq2 library (Love et al, 2014). Genes which were not expressed in at least two samples were filtered out prior to running the analysis.

Data from Jacobovitz et al (Jacobovitz et al, 2021; Data ref: Jacobovitz et al, 2021) were processed in the same manner as described above. The RNA-seq libraries of Symbiotic cells (SymSym: SRR10423689-94) were compared to the cells of aposymbiotic larvae (ApoApo: SRR10423679-83), aposymbiotic cells of symbiotic larvae (SymApo: SRR10423684-88), and cells hosting *M. gaditana* (Mgad: SRR10546813-15). The number of replicates required to keep a gene in the differential gene analyses was adapted in each case so that every gene kept in the analysis was expressed in at least $N$ samples with $N =$ (lower number of replicates across the two conditions) − 1.

### Statistics

For main figures, data were analyzed using Prism 9 (Version 9.1.1; GraphPad Software, LLC). For expanded view Figs. EV2 and EV4, figures were generated and analyzed using R Studio Version 2023.12.1 + 402 (2023.12.1 + 402). No blinded quantification was done for analysis.

The Shapiro test for normality was performed in R for all datasets. Descriptive statistics (mean and standard deviation) and SEM were calculated. When two parameters were compared, paired parametric $t$ tests were used to calculate $P$ values (significance level $P < 0.05$) using Prism Graphpad. When more than two parameters were analyzed, ANOVA was performed, and Tukey's multiple comparison tests were performed to determine statistical significance for each comparison (alpha = 0.05).

## Data availability

No original transcriptomics datasets were generated; only previously described publicly available datasets were reanalyzed, the details of which are in the respective "Methods" sections.

The source data of this paper are collected in the following database record: biostudies:S-SCDT-10_1038-S44319-025-00645-9.

## Peer review information

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

## Acknowledgements

We would like to acknowledge feedback and editing support from Benjamin Jenkins. Funding was provided by the H2020 European Research Council (ERC Consolidator Grant 724715) to AG, an EMBO and Alexander von Humboldt Postdoctoral fellowships to VSJ, and a Marie Curie Postdoctoral fellowship (101152260) to MF.

## Author contributions

**Victor A S Jones**: Conceptualization; Formal analysis; Methodology; Project administration. **Melanie Dörr**: Formal analysis; Methodology; Writing—original draft; Writing—review and editing. **Isabelle Siemers**: Methodology. **Sebastian Rupp**: Conceptualization; Formal analysis; Methodology; Writing—original draft; Project administration; Writing—review and editing. **Sami El Hilali**: Formal analysis. **Sara Brites**: Methodology. **Joachim M Surm**: Software; Formal analysis; Methodology. **Ira Maegele**: Methodology; Writing—original draft; Writing—review and editing. **Sebastian G Gornik**: Conceptualization; Resources; Software; Formal analysis; Funding acquisition; Methodology; Project administration; Writing—review and editing. **Meghan Ferguson**: Formal analysis; Writing—original draft; Writing—review and editing. **Annika Guse**: Conceptualization; Resources; Funding acquisition; Methodology; Project administration; Writing—review and editing.

Source data underlying figure panels in this paper may have individual authorship assigned. Where available, figure panel/source data authorship is listed in the following database record: biostudies:S-SCDT-10_1038-S44319-025-00645-9.

## Funding

## Disclosure and competing interests statement

VSJ is currently employed by biotech company Prolific Machines Inc., 6400 Hollis St, Emeryville, CA 94608, USA. The remaining authors declare no competing interests.

# Expanded View Figures

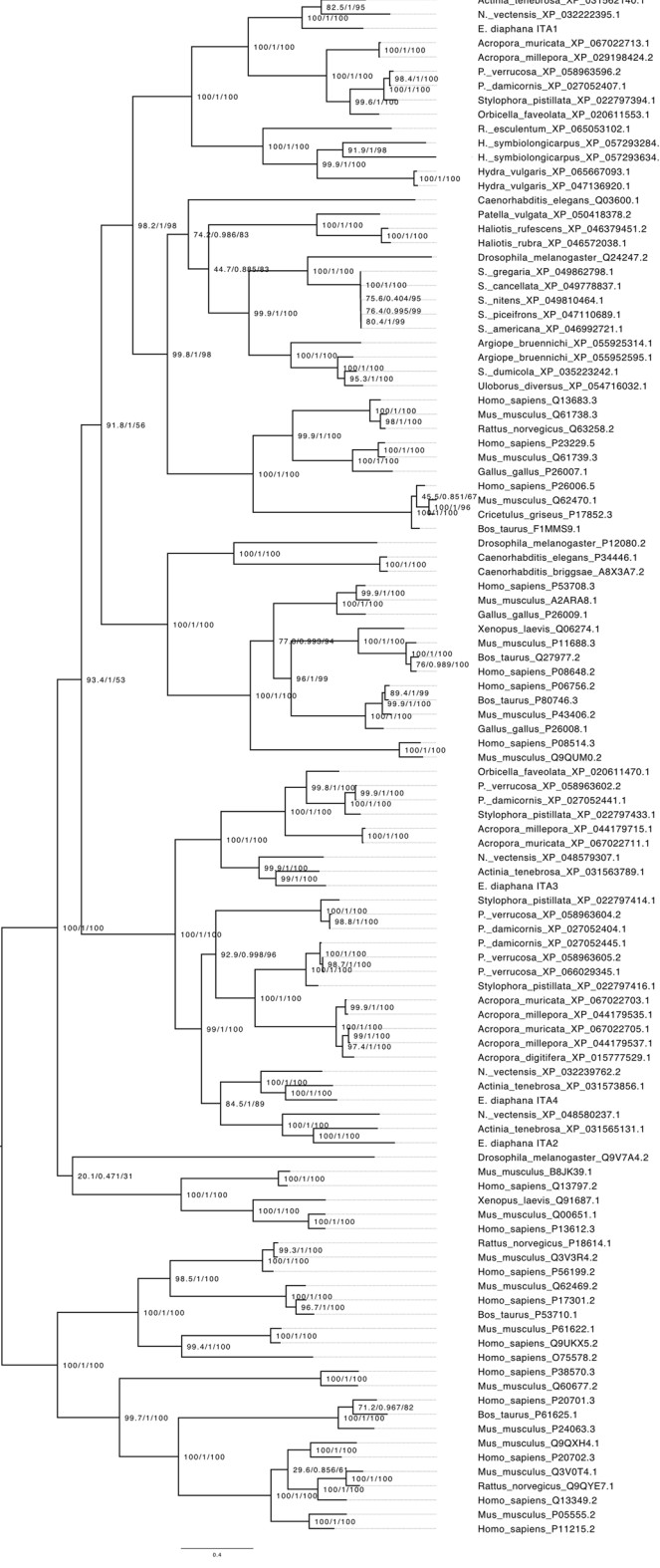

**Figure EV1.** Phylogenetic tree of integrin α proteins prior to collapse of subtrees and rotation of nodes.

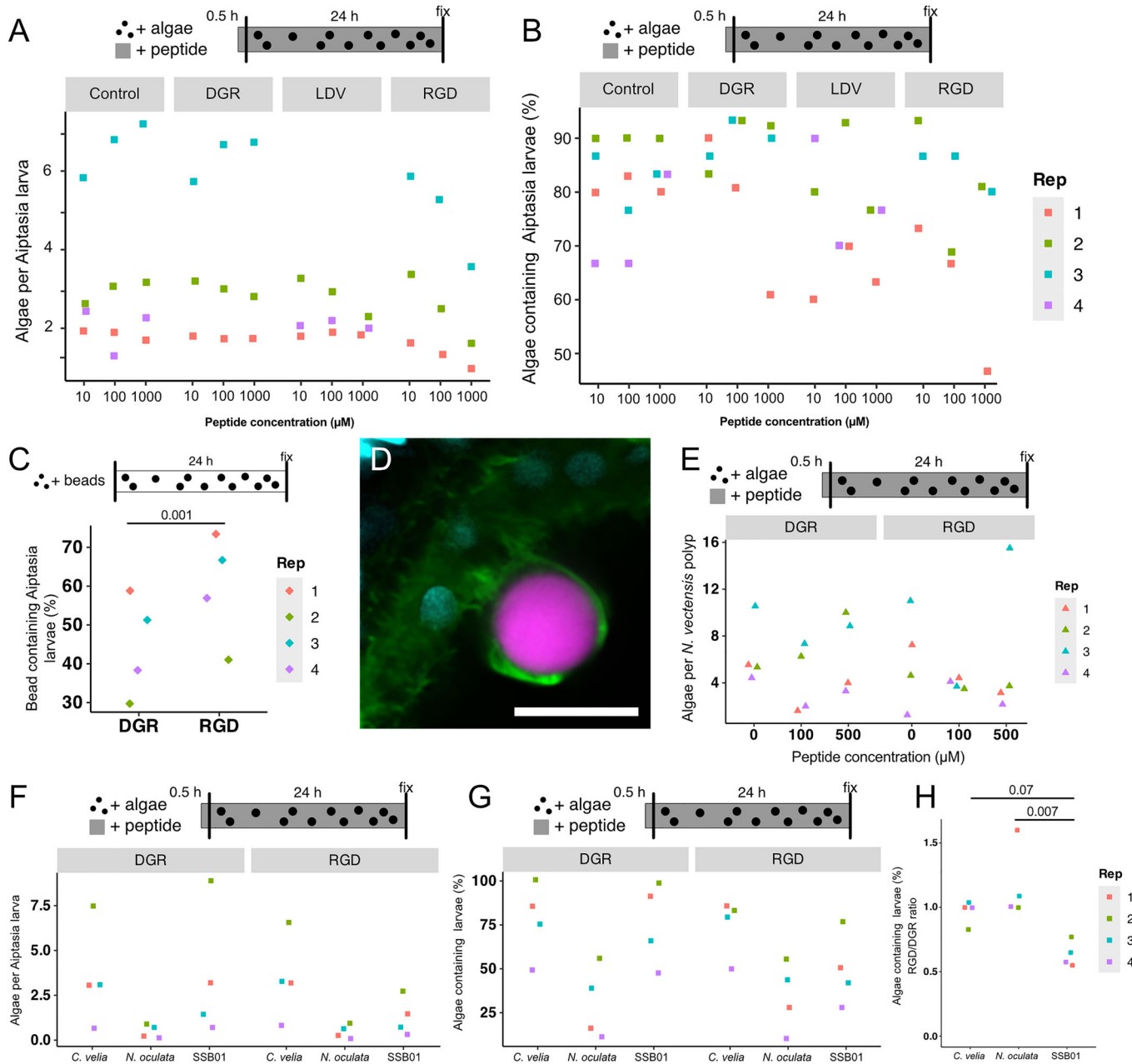

**Figure EV2. Raw and extended data for Fig. 3 RGD-integrins facilitate symbiont uptake in symbiotic cnidaria.**

(A, B) Aiptasia larvae were exposed to control peptide DGR, integrin ligand RGD or integrin ligand LDV for 30 min then exposed to symbionts (*B. minutum*, SSB01) for 24 h. $n = 3$. (A) SSB01 internalized by Aiptasia larvae. (B) Percent of Aiptasia larvae that phagocytosed SSB01. (C) Percent of Aiptasia larvae that phagocytosed peptide-coated beads. Beads were coated with either control peptide DGR or integrin ligand RGD then incubated with Aiptasia larvae for 24 h. $n = 4$. (D) Aiptasia larvae exposed to inert beads coated with RGD peptides and imaged after 48 h of exposure. Green = actin, Cyan = DNA, pink = bead. Scale bar = 10 μm. (E) SSB01 per *N. vectensis* polyp after a 24 h incubation and exposure to either the control peptide DGR or integrin ligand RGD. $n = 4$. (F) Algae per Aiptasia larvae and (G) Percent of Aiptasia larvae with algae inside upon RGD or DGR treatment (1000 μM peptide). $n = 4$. (H) Percent of larvae from G normalized by taking the ratio between RGD to DGR treatment. A ratio of less than 1 means RGD treatment has lower algae uptake than DGR. $n = 4$. For all plots, biological replicates are colored to highlight trends and natural variation between replicates. For (A, B, E–H) ANOVA was used to determine significance. Only significant differences are shown. For (C), significance was found via paired *t* test.

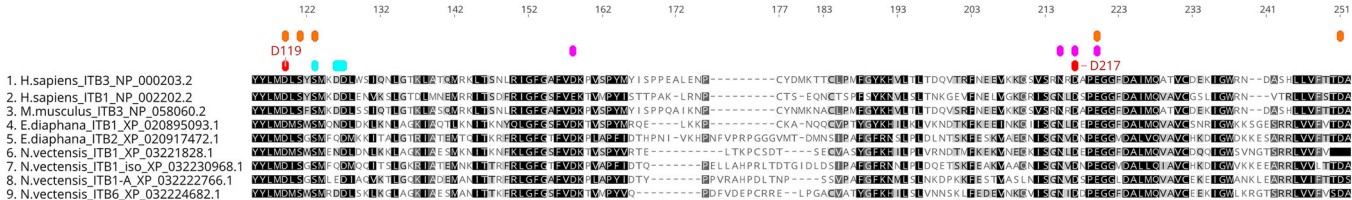

**Figure EV3.  Active sites of RGD-binding integrin and mutations used for this study.**

Highly conserved residues are highlighted in black and as conservation decreases this goes from gray to white. Conserved active sites are noted above with the colored labels. Orange = MIDAS. Turquoise = ADMIDAS. Pink = LIMBS. Red = used mutations to disrupt RGD binding.

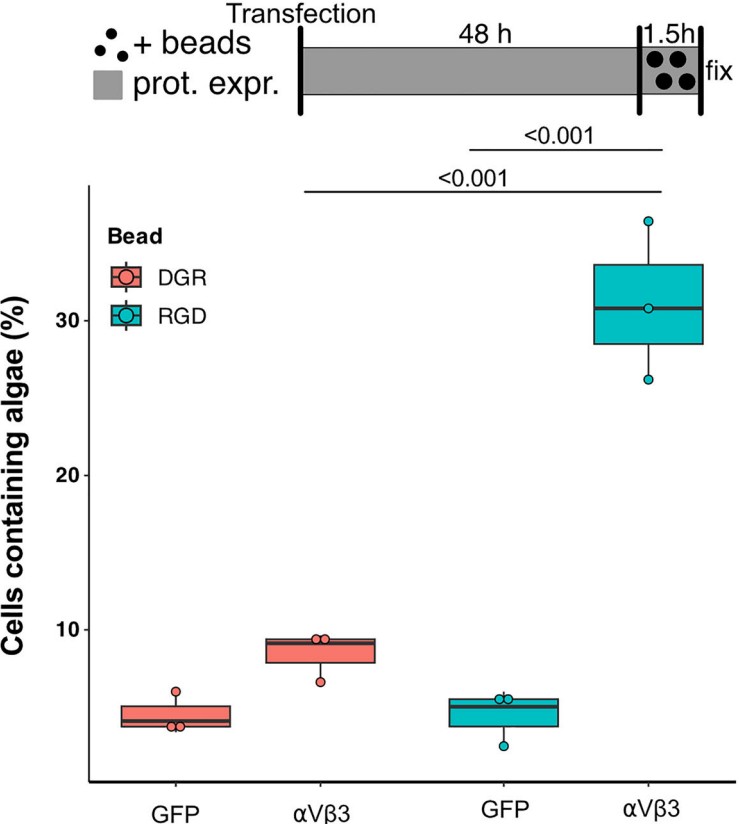

**Figure EV4.   Upon integrin overexpression, HEK cells preferentially phagocytose RGD-coated beads.**

HEK cells transfected with expression plasmids encoding mammalian integrin αV and β3 (each with halves of a split YFP and upon heterodimer formation fluoresce) and exposed to beads coated with either RGD or DGR. Percentage of HEK cells transfected with either αVβ3 integrins or GFP-CaaX as a control that phagocytosed RGD or DGR-coated beads. *n* = 3 biological replicates. Whiskers depict the minima and maxima, the center depicts the median and upper and lower edges of the box depicts the 25th and 75th percentile, respectively. Exact *P* values are 0.0000577 for the overexpression comparison and 0.0000173 for the RGD GFP and overexpression comparison. Statistical significance was found via ANOVA followed by Tukey.

