## [Peer Review File · EMBO Reports]

Integrins mediate symbiont-specific uptake in cnidarian larvae

Victor Jones, Melanie Dörr, Isabelle Siemers, Sebastian Rupp, Sami El Hilali, Sara Brites, Joachim Surm, Ira Maegele, Sebastian Gornik, Meghan Ferguson, and Annika Guse

Corresponding author(s): Annika Guse (annika.guse@biologie.uni-muenchen.de) , Meghan Ferguson (ferguson@bio.lmu.de)

Review Timeline:

Submission Date:	14th Feb 25
Editorial Decision:	27th Mar 25
Revision Received:	8th Aug 25
Editorial Decision:	13th Oct 25
Revision Received:	23rd Oct 25
Accepted:	10th Nov 25

Transaction Report:

Dear Dr. Guse

Thank you for the submission of your research manuscript to our journal. We have now received the full set of referee reports that is copied below.

As you will see, the referees acknowledge that the findings are interesting and that the conclusions are overall supported by the data presented but they also raise a number of concerns and have suggestions how to further strengthen the data that should be addressed.

Please note point 11 below on Data References, which would apply to reference (43) in your article.

Referee 2 and 3 both commented on statements based on "data not shown". Please note that our editorial policies also request that all data cited to support conclusions must be part of the manuscript.

Given these constructive comments, we would like to invite you to revise your manuscript with the understanding that the referee concerns (as detailed above and in their reports) must be fully addressed and their suggestions taken on board. Please address all referee concerns in a complete point-by-point response. Acceptance of the manuscript will depend on a positive outcome of a second round of review. It is EMBO Reports policy to allow a single round of revision only and acceptance or rejection of the manuscript will therefore depend on the completeness of your responses included in the next, final version of the manuscript.

We realize that it is difficult to revise to a specific deadline. In the interest of protecting the conceptual advance provided by the work, we recommend a revision within 3 months (June 27). Please discuss the revision progress ahead of this time with the editor if you require more time to complete the revisions.

I am also happy to discuss the revision further via e-mail or a video call, if you wish.

You can either publish the study as a short report or as a full article. For short reports, the revised manuscript should not exceed 27,000 characters (including spaces but excluding materials & methods and references) and 5 main plus 5 expanded view figures. The results and discussion sections must further be combined, which will help to shorten the manuscript text by eliminating some redundancy that is inevitable when discussing the same experiments twice.

For a normal article there are no length limitations, but it should have more than 5 main figures and the results and discussion sections must be separate. In both cases, the entire materials and methods must be included in the main manuscript file.

=====

IMPORTANT NOTE:

We perform an initial quality control of all revised manuscripts before re-review. Your manuscript will FAIL this control and the handling will be delayed IN CASE the following APPLIES:

- 1) A data availability section providing access to data deposited in public databases is missing. If you have not deposited any data, please add a sentence to the data availability section that explains that.
- 2) Your manuscript contains statistics and error bars based on $n=2$. Please use scatter blots in these cases. No statistics should be calculated if $n=2$.

=====

2) individual production quality figure files as .eps, .tif, .jpg (one file per figure).

Please download our Figure Preparation Guidelines (figure preparation pdf) from our Author Guidelines pages <https://www.embopress.org/page/journal/14693178/authorguide> for more info on how to prepare your figures.

4) a complete author checklist, which you can download from our author guidelines (). Please insert information in the checklist that is also reflected in the manuscript. The completed author checklist will also be part of the RPF.

5) Please note that all corresponding authors are required to supply an ORCID ID for their name upon submission of a revised manuscript (). Please find instructions on how to link your ORCID ID to your account in our manuscript tracking system in our Author guidelines
()

6) We replaced Supplementary Information with Expanded View (EV) Figures and Tables that are collapsible/expandable online. A maximum of 5 EV Figures can be typeset. EV Figures should be cited as 'Figure EV1, Figure EV2' etc... in the text and their respective legends should be included in the main text after the legends of regular figures.

7) Please note that a Data Availability section at the end of Materials and Methods is now mandatory. In case you have no data that requires deposition in a public database, please state so instead of refereeing to the database. See also < <https://www.embopress.org/page/journal/14693178/authorguide#dataavailability>>. Please note that the Data Availability Section is restricted to new primary data that are part of this study.

Additional information on source data and instruction on how to label the files are available .

10) Figure legends and data quantification:

- the name of the statistical test used to generate error bars and P values,
- the EXACT p-values,
- the number (n) of independent experiments (please specify technical or biological replicates) underlying each data point,
- the nature of the bars and error bars (s.d., s.e.m.)
- If the data are obtained from n {less than or equal to} 5, show the individual data points in addition to the SD or SEM.
- If the data are obtained from n {less than or equal to} 2, use scatter blots showing the individual data points.

11) Our journal encourages inclusion of *data citations in the reference list* to directly cite datasets that were re-used and obtained from public databases. Data citations in the article text are distinct from normal bibliographical citations and should directly link to the database records from which the data can be accessed. In the main text, data citations are formatted as follows: "Data ref: Smith et al, 2001" or "Data ref: NCBI Sequence Read Archive PRJNA342805, 2017". In the Reference list, data citations must be labeled with "[DATASET]". A data reference must provide the database name, accession

number/identifiers and a resolvable link to the landing page from which the data can be accessed at the end of the reference. Further instructions are available at .

12) All Materials and Methods need to be described in the main text using our 'Structured Methods' format. According to this format, the Methods section includes a Reagents and Tools Table (listing key reagents, experimental models, software and relevant equipment and including their sources and relevant identifiers) followed by a Methods and Protocols section describing the methods, ideally using a step-by-step protocol format. The aim is to facilitate adoption of the methodologies across labs. Please download and fill our Reagents and Tools Table template (.docx), which you can find in our author guidelines: <https://www.embopress.org/page/journal/14693178/authorguide#structuredmethods>.

13) As part of the EMBO publication's Transparent Editorial Process, EMBO Reports publishes online a Review Process File to accompany accepted manuscripts. This File will be published in conjunction with your paper and will include the referee reports, your point-by-point response and all pertinent correspondence relating to the manuscript.

Yours sincerely,

=====

Referee #1:

Summary

Jones et al. investigate the potential role of host integrins as mediators of specificity during early symbiosis establishment in the cnidarian model *Aiptasia*. Using previous transcriptomic data, they identify integrin-related genes as differentially expressed in symbiotic and aposymbiotic host cells. They uncover four relevant cnidarian integrin genes and resolve their phylogenetic relationships. They identify ITA1 as the most likely to bind RGD peptides and use FISH to show localized expression in the larval endoderm, where symbionts and host interact. They perform several blocking experiments in *Aiptasia* and coral larvae to demonstrate that when host integrins are competitively bound, symbiont uptake via phagocytosis is reduced. There is a strong element of specificity here, as only uptake of the homologous symbiont is reduced, whereas non-symbiotic algal uptake remains unaffected. Finally, they illustrate similar results in human embryonic kidney cells, where integrin overexpression resulted in enhanced symbiont uptake. These results indicate that integrin recognition is central to the uptake and/or retention of specific symbionts, though the authors rightly caution that it is not the only mechanism that regulates specificity.

This is a superb study and I only have minor notes for improvement. It explores a critical component of symbiosis establishment (molecular mechanisms of specificity) that has long been of interest to a broad community of researchers. It provides multiple lines of highly compelling evidence that integrins play a key role in symbiont-host recognition and therefore deserve more attention as MAMP-PRRs in mutualisms. Coral cell biologists have had a difficult time explaining how homologous partners maintain retain homologous symbionts during establishment even though heterologous symbionts as well as non-symbiotic algae can also enter host gastrodermal cells via phagocytosis. This work adds a new layer to the discussion and while I wouldn't call it "paradigm-shifting" (especially because it's been clear for a while that multiple signaling mechanisms must be at play), I do think it's a very exciting finding that will inspire many follow-up studies.

The experiments are technically impressive and well-executed (at least the ones I could evaluate-I know nothing about human cell line methodology), and the conclusions flow logically from the results. I was impressed with the writing, which was refreshingly concise. Relevant literature was incorporated throughout, with a good summary of previous findings and a very interesting description of how new results aligned with them. The work clearly fits with EMBO's criterion for a single key message with a substantial, novel, physiological insight. Excellent job.

Minor Notes

L46: "a new paradigm" is a tired phrase and I don't think it's appropriate, anyway. One could argue integrins are just another MAMP-PRR in the suite of molecular mechanisms we know are partially responsible for determining specificity (though it seems like a more consequential MAMP-PRR than many of the others that have previously been investigated).

L53: This is very minor, but I'd change to "can provide over 90% of the host's nutritional needs" (because the amount varies substantially depending on the association).

L75: I suggest changing to "Glycan-lectin" instead of "Lectin-glycan" to parallel the "MAMP-PRR" order in the next sentence (the glycan would be the MAMP and the lectin the PRR). Also, I feel that's the usual order, at least in the coral literature.

L117: Specify: "data from dissected larval endodermal cells" (I believe it's larval rather than adult based on Figure 1, correct?)

L127: Specify: "These data suggest that homologous symbionts specifically alter"

L383: Elsewhere in the manuscript (the caption to Figure 3), you indicate SSB01 is a strain of *Breviolum minutum*. I was pretty sure that strain SSB01 was confirmed to be *B. minutum* at some point, so I tracked it down. Baumgarten et al. (2015; PNAS) confirmed its identity via comparison of the Sym15 microsatellite flanker to the type specimen (strain rt002). Here's the relevant text from the supplement:

"To clarify the taxonomic position of SSB01, the microsatellite marker Sym15 was amplified and sequenced as described previously (55). Using BLASTN, the SSB01 Sym15 was found to align best to Sym15 from *S. minutum* (Clade B) genotypes FLAp2 (5) and rt002 (56) (E-values, 5e-100 and 4e-96, respectively), and less well to Sym15 from the sister species *S. psygmophilum* (56) (E-value, 9e-63). The sequence of the cp23S marker gene from SSB01 (54) is also highly similar to that of *S. minutum* (56). Maximum parsimony trees of the Sym15 and cp23S markers from various strains confirm that SSB01 represents a genotype of the species *S. minutum* (Fig. S1B)."

It's also a homologous symbiont of *Aiptasia*. Therefore, I'd change the text here in the methods as follows:

"*Breviolum minutum* (family Symbiodiniaceae, strain SSB01, homologous symbiont)"

I'd also change any instance of "*Breviolum* sp." to "*Breviolum minutum*" or "*B. minutum*" where appropriate.

Referee #2:

Summary

Referees are asked to supply answers to the following questions, with brief accompanying comments where appropriate:

1. Does this manuscript report a single key finding?

- YES. The key finding is that host integrins are involved in uptake/phagocytosis of potentially symbiotic but not other algae.

2. Is the reported work of significance (YES), or does it describe a confirmatory finding or one that has already been documented using other methods or in other organisms etc (NO)?

- YES. The manuscript is the first to show a role of integrins in cnidarian-dinoflagellate symbiosis establishment/specificity.

3. Is it of general interest to the molecular biology community?

If YES, please say why, in a single sentence. If NO, please state which more specialized community you feel it is aimed at (or none), in a single word or phrase.

- YES. The findings not only have implications for coral biologists and preservation/restoration under the thread of global warming but also highlight the evolutionary trajectories of such basic cellular mechanisms.

4. Is the single major finding robustly documented using independent lines of experimental evidence (YES), or is it really just a preliminary report requiring significant further data to become convincing, and thus more suited to a longer-format article (NO)?

- YES. The major finding is robustly supported by the experimental data with several different complimentary experimental approaches.

Free-form report:

The manuscript "Symbiont-specific uptake is mediated by integrins in cnidarian larvae" from Jones et al. explores the role of cnidarian RGD-integrins in the establishment/early stages of cnidarian-dinoflagellate symbiosis. After an initial bioinformatics approach to identify possible host integrins with a role in symbiosis, the authors employ a number of different model- and non-model potential host organisms to investigate the role of integrins in the uptake/retention of potential endosymbiont algae. They use peptides to block symbiont uptake as well as ligand-coated beads which are taken up by the host. The experiments show that potential symbionts, but not other algae rely at least in part on RGD-binding integrins to be phagocytosed by the potential host cells. This finding complements other reports which describe similar partial roles of glycan-lectin interactions on symbiont uptake. Overall, the manuscript shows convincing experimental data for the author's claims and should be of high interest for the coral symbiosis and restoration field but also for scientists studying the evolution of basic cellular recognition mechanisms and pathogen-host interactions. Yet, the authors might want to tone down some statements and thoroughly revise as well as shorten some sections of the text. Please see the following major and minor general as well as line-specific comments.

Major general comments:

1. Figure 1 shows differential expression of integrins identified in the *Aiptasia* genome. The one significantly differentially regulated one is "part of ITA3" while the full ITA3 transcript looks seems more ambiguous and even has a different homolog annotation ("integrin alpha-7" vs. "integrin alpha pat-2 isoform X1"). The same is true for ITA4 and "part of ITA4" (just less relevant since neither of them seems differentially expressed). This looks like the result of a mindless blast annotation algorithm and should be manually checked and presented in a more consistent way or at least be explained somewhere. With the current state of the *Aiptasia* genome, I would also suggest removing the homolog-isoform information from the figure and in return increase the font size of the remaining annotations slightly.
2. The reasoning for further investigation of ITA1 needs to be better explained. Figure 1 shows ITA3 (or "part of") as the only significantly differentially expressed alpha integrin, the next section looks at the phylogeny of the alpha units (why not also the beta subunits?) and has some information of the inferred ligands depending on the clades and then seemingly arbitrarily switches focus to the RGD-binding ITA1. The reason is kind of hidden away at the end of the second last paragraph of the introduction; I think a similar sentence should be written for the transition between the phylogeny and the expression localization. Another possibility could be to restructure the results part and first show the experiments on the ligands which clearly show that RGD but not LDV has an impact.
3. I think it would be good to show the non-normalized symbiont numbers per *Aiptasia* larva and the fraction of symbiotic vs. aposymbiotic larvae in a supplementary figure.
4. I find it quite confusing to refer to SSB01 as symbiont in all contexts. I would suggest to clearly separate symbiosis e.g., between *Aiptasia* or *Acropora* and SSB01 from the other cases where SSB01 (but also to a lesser extent the other algae) are most likely just phagocytosed by non-host organisms/cells. *Nematostella* might be an edge case (has that been studied in more detail somewhere?) but for the HEK cells, I think a functional symbiosis can be excluded; similarly, the compartment the algal cells are located after uptake should not be called a symbiosome. The same is true for calling the cells "infected".
5. The experiments use only one symbiodiniacean strain and the other non-symbiotic algae are only distantly related. Have the authors thought about comparing SSB01 with other Symbiodiniaceae which do not form stable symbioses with *Aiptasia*, e.g., strains SSA03 or SSE01 (see Tran et al., 2024)? I think experiments with those strains could prove very informative without adding too much length to the manuscript and could be done relatively easily.
6. I would recommend shortening and restructuring the discussion, at some parts there is excess, unnecessary detail and some of the paragraphs have some redundancies and repetitions. Especially the third paragraph could easily be cut to half length.
7. On several occasions, the authors seem to overstate the role of the integrins a bit. The data show a clear involvement in symbiont uptake, but I would not call it "crucial" or "key players".

Minor general comments:

1. The authors call the gastrodermal cell layer consistently endoderm. I think in the wider convention is that endo-, meso- and ectoderm are terms only used during the initial embryogenic development before final tissue differentiation.
2. Gene/transcript names should be italicized to distinguish them from the proteins.
3. There is some confusion and inconsistency regarding genes vs. proteins vs. transcripts in the first part of the results section (see specific comments below).
4. The *Nematostella*-specific sub figures in Figure 3 could be moved to a supplementary figure.
5. The authors should use the SI recommended unit symbol "h" for hour instead of "hr".
6. The Material and Methods section needs some careful revision in regards to consistent formatting of the units, e.g., spaces vs. no spaces, multiplication sign vs. upper case X or lower case x or just a space, etc.
7. A very minor detail but I would recommend using the proper minus sign character instead of the shorter hyphen. That makes it the same width as the plus sign and can improve readability when it comes to SSB01– vs. SSB01- in the text.

Line-specific comments:

L14: "...experimental data was performed here."

- Maybe "was obtained" or "was generated".

L31: "...RGD-binding integrins..."

- At this point, it is not clear that RGD just stands for a simple amino acid sequence/motif, I would change that to "Arginine-Glycine-Aspartic Acid".

L51: "...reef-building corals have acquired..."

- The "acquired" has unnecessary evolutionary implications which I would avoid.

L60: "...take up algae from their surroundings through horizontal (environmental) transmission..."

- The last part seems redundant, especially since "horizontal transmission" is not mentioned anywhere later in the manuscript.

L105: "...coral-algal symbiosis..."

- I would suggest writing "cnidarian-algal symbiosis" when not talking specifically ONLY about corals.

L106-109: "... (39-41)."

- These references are all only on *Nematostella*.

L110: "... , but not other algae,..."

- "... , but not that of other algae,..."

L118-119: "...compared their gene expression to endodermal cells from larvae that were exposed to, but did not contain symbionts (SSB01-)..."

- This description is a bit unclear; it implies that the SSB01– cells were from whole larvae that were exposed to algae but stayed aposymbiotic while it were actually (as far as I understand from ref. 43) these cells without algae were actually isolated from symbiotic larvae.

L122: "...,including one integrin alpha and one beta subunit,..."

- I would recommend to give the names of these two genes in the text.

L130-136: Legend Figure 1.

- I think, the two sources for the sequences, Lehnert et al., 2012 and Baumgarten et al., 2015, should be mentioned somewhere in the text, maybe together with a short reasoning why for ITA1, the sequence of the transcript from the older, transcriptomics data was used instead of the data from the newer genome paper and also what the exact meaning of the annotations "part of ITA3" and "part of ITA4" is and why there are conflicts between the descriptions of the full transcript and the partial transcripts.

L138-157:

- This section mixes genes, transcripts and proteins semi-randomly. I would recommend to keep it consistent and refer to the proteins in the first paragraph and to the transcripts for the in situ hybridization.

L143-144: "Cnidarian ITA Clade 1 contains ITA1 (NM_001434660.1, integrin alpha-6),..."

- The figure looks at the phylogeny at the protein level, yet the transcript identifier is listed here.

- See also previous comment on citing the references for the transcriptomic and genomic data and explaining the reason for using a mix of both.

L151: "...the alpha subunit ITA1 (NM_001434660.1, integrin alpha-6)..."

- Redundant, already described in L143.

L159: "Figure 2. Predicted RGD-binding *Aiptasia* Integrin alpha 1..."

- That should be lower case "integrin".

L164: "Numbers on Branches and scalebar indicate substitutions per site B-D) Localization..."

- "Numbers on branches and scale bar indicate substitutions per site. B-D) Localization..."

L170-L215:

- There are some errors in the spelling of the different peptides (see following comments), I would suggest only referring to the relevant tripeptide motifs after initial introduction of the different constructs (as done in the legend for Figure 3). That should make it easier for the reader and help spotting spelling errors.

L189: "...the control peptide SGDRG..."

- "SDGRG" ?

L207: "...Symbiodinaceae family..."

- "Symbiodiniaceae"

L212-213: "...the control peptide (GDGRS)..."

- "SDGRG" ?

L173-174: "We found that none of the peptides significantly affected the fraction of infected larvae (data not shown)."

- I recommend showing the data in a supplementary figure as per EMBO Reports policy.

L197-198: "While not naturally symbiotic, *N. vectensis* larvae (tentacle bud stage) can take up low amounts of both symbionts and non-symbiotic microalgae (Fig. 3I, J)."

- That is quite interesting, is there a publication on that behavior which could be referenced here?

L221: "...(*Breviolum minutum*, SSB01)".

- This is the only time the strain is described that way, at all other places it is described as *Breviolum* sp.

L232: "(L) Algae per larvae..."

- It should be mentioned that these are *Aiptasia* larvae again.

- It might be also a good idea to mention the concentration of the peptides, I assume it is 1000 μ M as for the *Acropora* larvae.

L244: "...compared to fGFP-expressing control cells..."

- I think this needs an explanation as it is not a standard annotation. I would suggest just calling it GFP-CaaX (I assume that is the c-terminal tag of the eGFP) because this is the actual encoded protein, and readers should be more familiar with it (the same is true for the legend of Figure 4). I would still add a half sentence on the expected localization.

L279-280: "D) Percentage of symbionts (SSB01) and non-symbionts (*N. oculata* and *C. velia*) infected HEK cells that express".

- That sentence needs some rephrasing, maybe start with the transgene expression as first criterium and then the number of those cells with phagocytosed algae.

L290: "...heterologous symbionts..."

- I think readers outside of the coral field could have difficulties understanding the meaning of this term without more context.

L298: "...(54)..."

- Should that not be reference 43?

L304-306: "...Bay et al. (55) obtained similar results when modifying symbiont surface molecules, and suggested that the observed reduction in uptake was more due to post-phagocytosis retention rather than pre-phagocytosis recognition (55),..."

- The first mention of the reference can be removed.

L376-379: "As such, *Aiptasia* positions itself as a simple yet powerful model organism to uncover the mechanisms of symbiosis establishment and could potentially provide key information for the generation of symbiont host pairings resistant to the challenges faced by coral reefs in today's changing climate."

- This sentence does not fit to the previous one.

L390: "...on a 12 hr light / 12 hr dark cycle under ~ 20 - 25 $\mu\text{mol m}^{-2} \text{s}^{-1}$ of photosynthetically active radiation (PAR),..."

- The authors should use the SI unit recommendation for hour; "h".

- There should be a space or midpoint between the units.

- Another information should be the light temperature/phosphor of the fluorescent tubes or LEDs (same for the *Aiptasia* and *Nematostella* maintenance).

L391: "..., 1 - 2 weeks post-splitting."

- I assume that that means that infections were performed 1-2 weeks after splitting the cultures but would suggest to add some clarifying words.

L457: "Infection with algae of beads were performed..."

- "Infections with algae or beads were performed..."

L460: "...7 % magnesium chloride hexahydrate..."

- I would suggest describing the molarity instead of the percentage, that would also allow shortening by removing the "hexahydrate".

L464: "...f-actin..."

- I think it is more common to capitalize the F; "F-actin".

L474: "...Hoechst..."

- "Hoechst 33258".

L529-536: "..."

- I would recommend reporting the quantities of the plasmids in pmol.

L532: "250 ng pEGFP-f (farnesylated eGFP, AGP57;..."

- Is that number correct or should it be 750 ng as well?

- As written previously, I would suggest to at least mention the actual genetically encoded tag (e.g., eGFP-CaaX).

L538: "...sterile Poly-L-lysine ..."

- "...sterile poly-L-lysine..."

L500-519: "..."

- I think these two sections could be significantly shortened and even combined into one general cloning paragraph as they both describe widely used standard techniques.

L555-560: "..."

- At least partially redundant with microscopy part on larval infections. Maybe merge both sections into a general microscopy section.

L568: "...T7 or Sp6 Polymerase..."

- "...T7 or SP6 polymerase..."

L563-610: "..."

- In the previous section PBS+Tween20 was abbreviated "PBS-T", here it is "PBT", I would suggest using only one abbreviation (even though the Tween20 concentrations are slightly different).

- In the previous sections, reagents were listed with their manufacturer number, I think this would help reproducibility of the in situ hybridization protocol as well.

L595: "...anti-Dig alkaline phosphates..."

- "anti-Dig alkaline phosphatase".

L630-631: "...refseq accession numbers XP_028518220.1, XP_028518220.1, 630 XP_020908361.1, and XP_020913056.1..."

- The first two accession numbers are identical, the first one (for ITA1) should be NP_001421589.1.

Referee #3:

How symbiosis forms in cnidarians is an important question for the general study of host-microbe interactions, particularly in cnidarians, due to their ecological importance. The manuscript identifies integrins as important selective molecules driving symbiosis formation in cnidarians. Together, the data suggests this is true. However, the data is often presented in a way that can obstruct the actual variation in experiments, sometimes lacks certain controls, and some implications are overly generalized. This manuscript can be improved and revised to help the reader and the field. I have summarized major and minor comments to help aid the manuscript.

Major comments:

Line 45: The manuscript often states that the data support the role of integrins in selecting symbiotic vs. non-symbiotic algae.

How Symbiodiniaceae species are selected is an important question in the field. However, the non-symbiotic strains of algae used in the study do not appear to be relevant to these claims because they do not naturally associate with any cnidarian (in my understanding), nor are they closely related to the symbiotic algae. The authors previously reported differences in specificity with strain SSB01, SSA01, and not SSE01 - which are feasible experiments. To make the claims in the paper, I feel the experiments need to be done with these other strains, which are much better suited to determine if integrins are important for selecting particular algae. Repeating these experiments is only important in Aiptasia (due to feasibility with coral in revision). Regardless, the text (introduction, significance, and discussion) needs to be adjusted to not overgeneralize the role of integrins in algal selection.

Line 120-123: I have a few questions about the RNAseq study. First, it is important to note that this data was already published in a previous manuscript. However, the genes shown in Fig. 1 may be a different subset than the previous paper. I am concerned about the general quality of sequencing data. The vast majority of genes are downregulated in symbiotic cells (in the original paper), the variance is high across replicates, the log-fold changes are extreme [replicates having +/- 10 or more differences in log expression, see Figure 1 row 1], and the differences shown are not statistically significant for the vast majority of the genes and comparisons shown. Together, this makes me concerned about interpreting these results. Are there simple explanations for these observations? There are several published Apo/Sym bulk RNAseq datasets published (in Aiptasia and coral). Are the genes associated with symbiosis (positive or negative) from these bulk RNAseq experiments differentially expressed in the current dataset (for example, the orthologs of NPC2)?

Figure 1 and 2: A central point in the paper is that integrins are dynamically regulated during symbiosis formation. Indeed, the discussion presents a model for why dynamic regulation is important for symbiosis control. However, due to the challenges with the RNAseq above, it is hard to tell if that differential regulation is accurate. I recommend orthogonal measurements of integrin expression to support these claims. qPCR of these genes in Apo/Sym larvae and adults and FISH in Apo/Sym larvae would be needed to support the model.

Figure 2: It is not clear why ITA1 was the focus. This gene does not appear to be differentially regulated in Figure 1. In situ of the other differentially expressed genes would support the proposed model.

Figure 2: The chromogenic and fluorescent in situ have different expression patterns. Why is that? Also, higher-resolution/magnification images of each help support determine any differences and the cell types involved. As discussed above, showing differential expression based on infection with mRNA localization is needed in the context of the RNAseq.

Line 173-174: Data not shown - would it be possible to show the data?

Line 177-179: A concern with drug and peptide treatment is off-target impacts. Is there orthogonal data that you can provide showing that the peptide should disrupt integrin functions in Aiptasia? An AlphaFold experiment with each of the integrins and each peptide used might help.

Figure 3-4: Showing only normalized data obscures the experiment variation. Providing raw data and unnormalized plots in the supplement will aid in reproducibility.

Figure 3-4: Statistics - I am not sure that ANOVAs and t-tests are appropriate tests for the normalized data because the data are not normally distributed, violating the test's assumptions. The data in Figure 4D have overlapping SE but are significant, which concerns me. Can the authors clarify?

Figure 4D: Is there a bead-only control? This control is important to know which peptide is having an impact on symbiosis.

Figure 3 F-H: How were algae determined to be inside cells verse in the gut cavity in the coral experiments?

Figure 4: Is the increase in algal infection specific to YFP-positive cells in A? If not, what is the explanation? Also, how many times was the experiment performed? (Sorry if I missed it).

Minor comments:

Introduction:

1. The introduction could be improved by discussing the evidence for and against pre- and post-phagocytic mechanisms while citing the appropriate literature.
2. Line 115-116: More details about this sentence in the introduction would help the reader frame the study.

Discussion:

Line 298-301: It has been observed that more than one alga can be in a gasterodermal cell. It is not known that only one alga can get into each cell. Can the authors explain this inference?

Line 369 - Xenia 'soft' coral. I think that will help the evolutionary biologists.

Reviewers comments are bolded, our response is in normal text.

Referee #1:

Summary

Jones et al. investigate the potential role of host integrins as mediators of specificity during early symbiosis establishment in the cnidarian model *Aiptasia*. Using previous transcriptomic data, they identify integrin-related genes as differentially expressed in symbiotic and aposymbiotic host cells. They uncover four relevant cnidarian integrin genes and resolve their phylogenetic relationships. They identify ITA1 as the most likely to bind RGD peptides and use FISH to show localized expression in the larval endoderm, where symbionts and host interact. They perform several blocking experiments in *Aiptasia* and coral larvae to demonstrate that when host integrins are competitively bound, symbiont uptake via phagocytosis is reduced. There is a strong element of specificity here, as only uptake of the homologous symbiont is reduced, whereas non-symbiotic algal uptake remains unaffected. Finally, they illustrate similar results in human embryonic kidney cells, where integrin overexpression resulted in enhanced symbiont uptake. These results indicate that integrin recognition is central to the uptake and/or retention of specific symbionts, though the authors rightly caution that it is not the only mechanism that regulates specificity.

This is a superb study and I only have minor notes for improvement. It explores a critical component of symbiosis establishment (molecular mechanisms of specificity) that has long been of interest to a broad community of researchers. It provides multiple lines of highly compelling evidence that integrins play a key role in symbiont-host recognition and therefore deserve more attention as MAMP-PRRs in mutualisms. Coral cell biologists have had a difficult time explaining how homologous partners maintain retain homologous symbionts during establishment even though heterologous symbionts as well as non-symbiotic algae can also enter host gastrodermal cells via phagocytosis. This work adds a new layer to the discussion and while I wouldn't call it "paradigm-shifting" (especially because it's been clear for a while that multiple signaling mechanisms must be at play), I do think it's a very exciting finding that will inspire many follow-up studies.

The experiments are technically impressive and well-executed (at least the ones I could evaluate-I know nothing about human cell line methodology), and the conclusions flow logically from the results. I was impressed with the writing, which was refreshingly concise. Relevant literature was incorporated throughout, with a good summary of previous findings and a very interesting description of how new results aligned with them. The work clearly fits with EMBO's criterion for a single key message with a substantial, novel, physiological insight. Excellent job.

Minor Notes

L46: "a new paradigm" is a tired phrase and I don't think it's appropriate, anyway. One could argue integrins are just another MAMP-PRR in the suite of molecular mechanisms we know are partially responsible for determining specificity (though it seems like a more consequential MAMP-PRR than many of the others that have previously been investigated).

We have changed the wording to "novel mechanism".

L53: This is very minor, but I'd change to "can provide over 90% of the host's nutritional needs" (because the amount varies substantially depending on the association).

We have updated this.

L75: I suggest changing to "Glycan-lectin" instead of "Lectin-glycan" to parallel the "MAMP-PRR" order in the next sentence (the glycan would be the MAMP and the lectin the PRR). Also, I feel that's the usual order, at least in the coral literature.

We have updated this.

L117: Specify: "data from dissected larval endodermal cells" (I believe it's larval rather than adult based on Figure 1, correct?)

Yes, it is from larvae, we have updated this.

L127: Specify: "These data suggest that homologous symbionts specifically alter"

We have updated this.

L383: Elsewhere in the manuscript (the caption to Figure 3), you indicate SSB01 is a strain of *Breviolum minutum*. I was pretty sure that strain SSB01 was confirmed to be *B. minutum* at some point, so I tracked it down. Baumgarten et al. (2015; PNAS) confirmed its identity via comparison of the *Sym15* microsatellite flanker to the type specimen (strain rto02). Here's the relevant text from the supplement:

"To clarify the taxonomic position of SSB01, the microsatellite marker Sym15 was amplified and sequenced as described previously (55). Using BLASTN, the SSB01 Sym15 was found to align best to Sym15 from *S. minutum* (Clade B) genotypes FLAP2 (5) and rtoo2 (56) (E-values, 5e-100 and 4e-96, respectively), and less well to Sym15 from the sister species *S. psymophilum* (56) (E-value, 9e-63). The sequence of the cp23S marker gene from SSB01 (54) is also highly similar to that of *S. minutum* (56). Maximum parsimony trees of the Sym15 and cp23S markers from various strains confirm that SSB01 represents a genotype of the species *S. minutum* (Fig. S1B)."

It's also a homologous symbiont of *Aiptasia*. Therefore, I'd change the text here in the methods as follows:

"*Breviolum minutum* (family Symbiodiniaceae, strain SSB01, homologous symbiont)"

I'd also change any instance of "*Breviolum* sp." to "*Breviolum minutum*" or "*B. minutum*" where appropriate. We have updated this throughout the text.

Referee #2:

Summary

The manuscript "Symbiont-specific uptake is mediated by integrins in cnidarian larvae" from Jones et al. explores the role of cnidarian RGD-integrins in the establishment/early stages of cnidarian-dinoflagellate symbiosis. After an initial bioinformatics approach to identify possible host integrins with a role in symbiosis, the authors employ a number of different model- and non-model potential host organisms to investigate the role of integrins in the uptake/retention of potential endosymbiont algae. They use peptides to block symbiont uptake as well as ligand-coated beads which are taken up by the host. The experiments show that potential symbionts, but not other algae rely at least in part on RGD-binding integrins to be phagocytosed by the potential host cells. This finding complements other reports which describe similar partial roles of glycan-lectin interactions on symbiont uptake. Overall, the manuscript shows convincing experimental data for the author's claims and should be of high interest for the coral symbiosis and restoration field but also for scientists studying the evolution of basic cellular recognition mechanisms and pathogen-host interactions. Yet, the authors might want to tone down some statements and thoroughly revise as well as shorten some sections of the text. Please see the following major and minor general as well as line-specific comments.

Major general comments:

1. Figure 1 shows differential expression of integrins identified in the *Aiptasia* genome. The one significantly differentially regulated one is "part of ITA3" while the full ITA3 transcript looks seems more ambiguous and even has a different homolog annotation ("integrin alpha-7" vs. "integrin alpha pat-2 isoform X1"). The same is true for ITA4 and "part of ITA4" (just less relevant since neither of them seems differentially expressed). This looks like the result of a mindless blast annotation algorithm and should be manually checked and presented in a more consistent way or at least be explained somewhere. With the current state of the *Aiptasia* genome, I would also suggest removing the homolog-isoform information from the figure and in return increase the font size of the remaining annotations slightly.

We agree that the annotations for the integrins were confusing, mostly because of using gene models of insufficient quality. We now revised the gene models and generated curated gene models based on the set of initial gene models submitted to genbank for three integrins. We also analyzed the protein domains of the predicted proteins to ensure that our analyses were based on transcripts encoding for peptides that look complete and hypothetically functional. This analysis is now described in the result and the method sections of the revised manuscript. To further reduce confusion, we repeated the analysis with counts at the level of genes, excluding the isoform information which is not relevant for our study.

2. The reasoning for further investigation of ITA1 needs to be better explained. Figure 1 shows ITA3 (or "part of") as the only significantly differentially expressed alpha integrin, the next section looks at the phylogeny of the alpha units (why not also the beta subunits?) and has some information of the inferred ligands depending on the clades and then seemingly arbitrarily switches focus to the RGD-binding ITA1. The reason is kind of hidden away at the end of the second last paragraph of the introduction; I think a similar sentence should be written for the transition between the phylogeny and the expression localization. Another possibility could be to restructure the results part and first show the experiments on the ligands which clearly show that RGD but not LDV has an impact.

We have updated the text accordingly to make our focus on ITA1 clearer, including referencing that alpha integrins dictate ligand specificity. Indeed, in our transcriptomics experiments, ITA1 is downregulated, albeit not significantly. However, given that based on our phylogeny it is the only RGD-binding integrin in *Aiptasia*, we speculated that in analogy to various human pathogens which have co-opted RGD-binding integrin receptors to adhere to, enter, enhance colonization and replication, and spread within the host, ITA1 might play a role in symbiont uptake.

3. I think it would be good to show the non-normalized symbiont numbers per Aiptasia larva and the fraction of symbiotic vs. aposymbiotic larvae in a supplementary figure.

We have added figures for the raw non-normalized data for figure 3B, 3K, 3L in the supplement.

4. I find it quite confusing to refer to SSB01 as symbiont in all contexts. I would suggest to clearly separate symbiosis e.g., between Aiptasia or Acropora and SSB01 from the other cases where SSB01 (but also to a lesser extent the other algae) are most likely just phagocytosed by non-host organisms/cells. Nematostella might be an edge case (has that been studied in more detail somewhere?) but for the HEK cells, I think a functional symbiosis can be excluded; similarly, the compartment the algal cells are located after uptake should not be called a symbiosome. The same is true for calling the cells "infected".

We have changed this throughout the text to distinguish more clearly between symbiont versus non symbionts and refrain from using the term infection.

5. The experiments use only one symbiodiniacean strain and the other non-symbiotic algae are only distantly related. Have the authors thought about comparing SSB01 with other Symbiodiniaceae which do not form stable symbioses with Aiptasia, e.g., strains SSA03 or SSE01 (see Tran et al., 2024)? I think experiments with those strains could prove very informative without adding too much length to the manuscript and could be done relatively easily.

Yes, we agree that these experiments will be important to understand the specificity of the integrin-dinoflagellate interactions in molecular detail. However, this particular study focused in analogy to our previous comparative approaches mostly on the host side (Jacobovitz et al., 2021, Nat. Microbio). We therefore again used three distinct classes of algae: a true symbiont, a closely related microalga (*C. velia*) that is naturally establishing an intracellular partnership with corals while generally considered not mutualistic, and a distantly related marine alga (*N. oculata*) representing a random alga that could be encountered in the wild while on the search for symbionts, as well as an inert particle/non-living object. We therefore believe that our side-by-side comparison is actually quite comprehensive and providing substantially new information (e.g. evidence for a key receptor directly recognizing a symbiont).

Moreover, in separate projects, we are currently probing the influence of distinct symbionts on the host cell biology and we are elucidating the role of integrins in symbiosis establishment in more depth, including determining which exact integrin pairs are responsible for enhancing symbiont uptake, whether integrin signalling is involved in symbiont maintenance, what the ligand is that these integrins are detecting and whether differences in dinoflagellate uptake is due to differences in ligand-integrin binding kinetics.

Therefore, we hope that the reviewer agrees with us that these experiments while indeed very interesting is beyond the scope of this foundational paper that bring integrins onto the map for symbiont uptake and that such experiments deserve a detailed molecular undertaking from multiple angles to fully appreciate the mechanisms behind integrin mediated symbiont selection and specificity.

6. I would recommend shortening and restructuring the discussion, at some parts there is excess, unnecessary detail and some of the paragraphs have some redundancies and repetitions. Especially the third paragraph could easily be cut to half length.

As per EMBO short report style formatting we have shortened the text and integrated the discussion into the results section.

7. On several occasions, the authors seem to overstate the role of the integrins a bit. The data show a clear involvement in symbiont uptake, but I would not call it "crucial" or "key players".

We have gone through the text and removed these terms and give credit but not to overstate the role of integrins.

Minor general comments:

1. The authors call the gastrodermal cell layer consistently endoderm. I think in the wider convention is that endo-, meso- and ectoderm are terms only used during the initial embryogenic development before final tissue differentiation.

We have changed all instances of endoderm to gastroderm.

2. Gene/transcript names should be italicized to distinguish them from the proteins.

We have gone through the text and figures and italicized any instances of gene names.

3. There is some confusion and inconsistency regarding genes vs. proteins vs. transcripts in the first part of the results section (see specific comments below).

4. The Nematostella-specific sub figures in Figure 3 could be moved to a supplementary figure.

We have decided to keep these figures in the main figure since we believe their findings are an important feature of this manuscript.

5. The authors should use the SI recommended unit symbol "h" for hour instead of "hr".

We have changed all instances of hr to h.

6. The Material and Methods section needs some careful revision in regards to consistent formatting of the units, e.g., spaces vs. no spaces, multiplication sign vs. upper case X or lower case x or just a space, etc.

We have reformatted the section to be consistent throughout.

7. A very minor detail but I would recommend using the proper minus sign character instead of the shorter hyphen. That makes it the same width as the plus sign and can improve readability when it comes to SSB01- vs. SSB01- in the text.

When updating figure 1 we have changed the figure so the SSB01- and SSB01+ labels are no longer present. Instead, we use SymSym to denote cells containing the symbiont SSB01 and SymApo to represent cells that do not contain SSB01, but that came from larvae where indeed other cells did contain SSB01. We hope that this updated labelling is more clear.

Line-specific comments:

L14: "...experimental data was performed here."

- Maybe "was obtained" or "was generated".

Changed this

L31: "...RGD-binding integrins..."

- At this point, it is not clear that RGD just stands for a simple amino acid sequence/motif, I would change that to "Arginine-Glycine-Aspartic Acid".

Changed abstract to talk of integrins more generally and took out instances of RGD

L51: "...reef-building corals have acquired..."

- The "acquired" has unnecessary evolutionary implications which I would avoid.

Changed this

L60: "...take up algae from their surroundings through horizontal (environmental) transmission..."

- The last part seems redundant, especially since "horizontal transmission" is not mentioned anywhere later in the manuscript.

Changed this

L105: "...coral-algal symbiosis..."

- I would suggest writing "cnidarian-algal symbiosis" when not talking specifically ONLY about corals.

Changed this

L106-109: "... (39-41)."

- These references are all only on *Nematostella*.

This sentence merely states the models that are used in the study, so these references have been removed since a more detailed description exists in the methods section.

L110: "..., but not other algae,..."

- "..., but not that of other algae,..."

Changed this

L118-119: "...compared their gene expression to endodermal cells from larvae that were exposed to, but did not contain symbionts (SSB01-)..."

- This description is a bit unclear; it implies that the SSB01- cells were from whole larvae that were exposed to algae but stayed aposymbiotic while it were actually (as far as I understand from ref. 43) these cells without algae were actually isolated from symbiotic larvae.

Changed this to be more clear

L122: "...,including one integrin alpha and one beta subunit,..."

- I would recommend to give the names of these two genes in the text.

From our updated transcriptomics analysis with the curated integrin gene models we now find that ITB2 is significantly downregulated in all comparisons where the individual gastrodermal cells were sequenced and we note this in the results text while referring to the specific genes.

L130-136: Legend Figure 1.

- I think, the two sources for the sequences, Lehnert et al., 2012 and Baumgarten et al., 2015, should be mentioned somewhere in the text, maybe together with a short reasoning why for ITA1, the sequence of the transcript from the older, transcriptomics data was used instead of the data from the newer genome paper and also what the exact meaning of the annotations "part of ITA3" and "part of ITA4" is and why there are conflicts between the descriptions of the full transcript and the partial transcripts.

We have updated the methods to describe how we have reanalyzed and manually curated these integrins in question.

Briefly, ITA4 on refseq was incomplete as it did not contain a signal peptide, while the sequences submitted by KAUST did contain a signal peptide. We therefore manually curated this for our analysis. ITA3 was split into two genes which lay next to each other on the genome. The shorter of the two contained the signal peptide, which was missing from the protein

domains from the longer transcript. So this was manually curated into a single gene. The "part of ITA₃" that we had in the original manuscript seems, under closer inspection, to be on a very small contig and likely an assembly error after comparing to other available genomes. Therefore, we ignored this gene for our updated analysis.

L138-157:

- This section mixes genes, transcripts and proteins semi-randomly. I would recommend to keep it consistent and refer to the proteins in the first paragraph and to the transcripts for the in situ hybridization.

We have updated the text to be as clear as possible here as to whether we are referring to proteins or genes.

L143-144: "Cnidarian ITA Clade 1 contains ITA1 (NM_001434660.1, integrin alpha-6),..."

- The figure looks at the phylogeny at the protein level, yet the transcript identifier is listed here.

- See also previous comment on citing the references for the transcriptomic and genomic data and explaining the reason for using a mix of both.

We have updated the text to use the protein identifier instead of the gene ID.

L151: "...the alpha subunit ITA1 (NM_001434660.1, integrin alpha-6)..."

- Redundant, already described in L143.

Deleted this redundancy

L159: "Figure 2. Predicted RGD-binding Aiptasia Integrin alpha 1..."

- That should be lower case "integrin".

Changed this

L164: "Numbers on Branches and scalebar indicate substitutions per site B-D) Localization..."

- "Numbers on branches and scale bar indicate substitutions per site. B-D) Localization..."

Changed this.

L170-L215:

- There are some errors in the spelling of the different peptides (see following comments), I would suggest only referring to the relevant tripeptide motifs after initial introduction of the different constructs (as done in the legend for Figure 3). That should make it easier for the reader and help spotting spelling errors.

We have updated this so that in the figure and text we just use the tripeptide motif but in the methods section describe the entire peptide used.

L189: "...the control peptide SGDRG..."

- "SDGRG" ?

Changed this

L207: "...Symbiodinaceae family..."

- "Symbiodiniaceae"

Changed this.

L212-213: "...the control peptide (GDGRS)..."

- "SDGRG" ?

Changed this

L173-174: "We found that none of the peptides significantly affected the fraction of infected larvae (data not shown)."

- I recommend showing the data in a supplementary figure as per EMBO Reports policy.

Yes we have added these data to the supplement.

L197-198: "While not naturally symbiotic, *N. vectensis* larvae (tentacle bud stage) can take up low amounts of both symbionts and non-symbiotic microalgae (Fig. 3I, J)."

- That is quite interesting, is there a publication on that behavior which could be referenced here?

These are observations from our lab that, to our knowledge, has not been published before.

L221: "...(*Breviolum minutum*, SSB01)".

- This is the only time the strain is described that way, at all other places it is described as *Breviolum* sp.

We have changed all instances of *Breviolum* sp. To *B. minutum* but generally stick to using SSB01 throughout the text.

L232: "(L) Algae per larvae..."

- It should be mentioned that these are *Aiptasia* larvae again.

- It might be also a good idea to mention the concentration of the peptides, I assume it is 1000 μ M as for the *Acropora* larvae.

We have made these changes.

L244: "...compared to fGFP-expressing control cells..."

- I think this needs an explanation as it is not a standard annotation. I would suggest just calling it GFP-CaaX (I assume that is the c-terminal tag of the eGFP) because this is the actual encoded protein, and readers should be more familiar with it (the same is true for the legend of Figure 4). I would still add a half sentence on the expected localization.

Correct, this is the c terminal CaaX tag of GFP. We have made these changes.

L279-280: "(D) Percentage of symbionts (SSBo1) and non-symbionts (N. oculata and C. velia) infected HEK cells that express".

- That sentence needs some rephrasing, maybe start with the transgene expression as first criterium and then the number of those cells with phagocytosed algae.

Changed this.

L290: "...heterologous symbionts...".

- I think readers outside of the coral field could have difficulties understanding the meaning of this term without more context.

Added a short explanation here.

L298: "...(54)..."

- Should that not be reference 43?

This is correct and we changed it.

L304-306: "...Bay et al. (55) obtained similar results when modifying symbiont surface molecules, and suggested that the observed reduction in uptake was more due to post-phagocytosis retention rather than pre-phagocytosis recognition (55),..."

- The first mention of the reference can be removed.

Changed this.

L376-379: "As such, Aiptasia positions itself as a simple yet powerful model organism to uncover the mechanisms of symbiosis establishment and could potentially provide key information for the generation of symbiont host pairings resistant to the challenges faced by coral reefs in today's changing climate."

- This sentence does not fit to the previous one.

We removed this sentence.

L390: "...on a 12 hr light / 12 hr dark cycle under ~ 20 - 25 $\mu\text{mol m}^{-2} \text{s}^{-1}$ of photosynthetically active radiation (PAR),..."

- The authors should use the SI unit recommendation for hour; "h".

- There should be a space or midpoint between the units.

- Another information should be the light temperature/phosphor of the fluorescent tubes or LEDs (same for the Aiptasia and Nematostella maintenance).

We have changed the units formatting throughout to be consistent and updated the methods with the light temperature.

L391: "..., 1 - 2 weeks post-splitting."

- I assume that that means that infections were performed 1-2 weeks after splitting the cultures but would suggest to add some clarifying words.

Changed this.

L457: "Infection with algae of beads were performed...".

- "Infections with algae or beads were performed...".

Changed this.

L460: "...7 % magnesium chloride hexahydrate...".

- I would suggest describing the molarity instead of the percentage, that would also allow shortening by removing the "hexahydrate".

We have updated this

L464: "...f-actin...".

- I think it is more common to capitalize the F; "F-actin".

Changed this.

L474: "...Hoechst...".

- "Hoechst 33258".

Changed this.

L529-536: "..."

- I would recommend reporting the quantities of the plasmids in pmol.

L532: "250 ng pEGFP-f (farnesylated eGFP, AGP57;...".

- Is that number correct or should it be 750 ng as well?

We have updated this to instead show the moles used. Indeed, it is correct we used less of the control plasmid, as the expression level is much more efficient with this plasmid. However, we also combine it with an empty vector to have a similar total quantity of DNA added to all conditions.

- As written previously, I would suggest to at least mention the actual genetically encoded tag (e.g., eGFP-CaaX).

L538: "...sterile Poly-L-lysine ...".

- "...sterile poly-L-lysine..."

Changed this.

L500-519: "...".

- I think these two sections could be significantly shortened and even combined into one general cloning paragraph as they both describe widely used standard techniques.

L555-560: "...".

- At least partially redundant with microscopy part on larval infections. Maybe merge both sections into a general microscopy section.

For the methods sections stated above we decided to keep the sections separate so methods are easier to distinguish between our different analyses.

L568: "...T7 or Sp6 Polymerase...".

- "...T7 or SP6 polymerase..."

Changed this.

L563-610: "...".

- In the previous section PBS+Tween20 was abbreviated "PBS-T", here it is "PBT", I would suggest using only one abbreviation (even though the Tween20 concentrations are slightly different).

- In the previous sections, reagents were listed with their manufacturer number, I think this would help reproducibility of the in situ hybridization protocol as well.

We have changed PBT to PBS-T. We have also updated the methods to include the information for the reagents used for the in situ.

L595: "...anti-Dig alkaline phosphates...".

- "anti-Dig alkaline phosphatase".

Changed this.

L630-631: "...refseq accession numbers XP_0285182201.1, XP_028518220.1, 630 XP_020908361.1, and XP_020913056.1...".

- The first two accession numbers are identical, the first one (for ITA1) should be NP_001421589.1.

Changed this.

Referee #3:

How symbiosis forms in cnidarians is an important question for the general study of host-microbe interactions, particularly in cnidarians, due to their ecological importance. The manuscript identifies integrins as important selective molecules driving symbiosis formation in cnidarians. Together, the data suggests this is true. However, the data is often presented in a way that can obstruct the actual variation in experiments, sometimes lacks certain controls, and some implications are overly generalized. This manuscript can be improved and revised to help the reader and the field. I have summarized major and minor comments to help aid the manuscript.

Major comments:

Line 45: The manuscript often states that the data support the role of integrins in selecting symbiotic vs. non-symbiotic algae. How Symbiodiniaceae species are selected is an important question in the field. However, the non-symbiotic strains of algae used in the study do not appear to be relevant to these claims because they do not naturally associate with any cnidarian (in my understanding), nor are they closely related to the symbiotic algae. The authors previously reported differences in specificity with strain SSB01, SSA01, and not SSE01 - which are feasible experiments. To make the claims in the paper, I feel the experiments need to be done with these other strains, which are much better suited to determine if integrins are important for selecting particular algae. Repeating these experiments is only important in Aiptasia (due to feasibility with coral in revision). Regardless, the text (introduction, significance, and discussion) needs to be adjusted to not overgeneralize the role of integrins in algal selection.

Yes, we agree that these experiments will be important to understand the specificity of the integrin-dinoflagellate interactions in molecular detail. However, this particular study focused in analogy to our previous comparative approaches mostly on the host side (Jacobovitz et al., 2021, Nat. Microbio). We therefore again used three distinct classes of algae: a true symbiont, a closely related microalga (*C. velia*) that is naturally establishing an intracellular partnership with corals while generally considered not mutualistic, and a distantly related marine alga (*N. oculata*) representing a random alga that could be encountered in the wild while on the search for symbionts, as well as an inert particle/non-living object. We therefore believe that our side-by-side comparison is actually quite comprehensive and providing substantially new information (e.g. evidence for a key receptor directly recognizing a symbiont).

Moreover, in separate projects, we are currently probing the influence of distinct symbionts on the host cell biology and we are elucidating the role of integrins in symbiosis establishment in more depth, including determining which exact integrin pairs are responsible for enhancing symbiont uptake, whether integrin signalling is involved in symbiont maintenance, what the ligand is that these integrins are detecting and whether differences in dinoflagellate uptake is due to differences in ligand-integrin binding kinetics.

Therefore, we hope that the reviewer agrees with us that these experiments while indeed very interesting is beyond the scope of this foundational paper that bring integrins onto the map for symbiont uptake and that such experiments deserve a detailed molecular undertaking from multiple angles to fully appreciate the mechanisms behind integrin mediated symbiont selection and specificity.

We have also gone through the text and attempted to not over generalize that integrin dependent uptake is specific to symbionts, but rather try to be explicit in stating that it is a mechanism for the uptake of SSB01.

Line 120-123: I have a few questions about the RNAseq study. First, it is important to note that this data was already published in a previous manuscript. However, the genes shown in Fig. 1 may be a different subset than the previous paper. I am concerned about the general quality of sequencing data. The vast majority of genes are downregulated in symbiotic cells (in the original paper), the variance is high across replicates, the log-fold changes are extreme [replicates having +/- 10 or more differences in log expression, see Figure 1 row 1], and the differences shown are not statistically significant for the vast majority of the genes and comparisons shown. Together, this makes me concerned about interpreting these results. Are there simple explanations for these observations? There are several published Apo/Sym bulk RNAseq datasets published (in Aiptasia and coral). Are the genes associated with symbiosis (positive or negative) from these bulk RNAseq experiments differentially expressed in the current dataset (for example, the orthologs of NPC2)?

Thanks for these detailed comments. We have taken multiple steps to address those comprehensively and further improve the manuscript:

- (1) We have now completely reworked the analysis for the RNA-seq shown in Figure 1 and included an additional RNA-Seq dataset from our lab to cross-validate the results shown. This includes the gene expression for the known symbiosis related NPC genes, which are upregulated in our datasets.
- (2) We have completely restructured and rewritten this part of the manuscript for improved clarity.
- (3) We have added an extra integrin protein domain analysis after carefully curating the genes by hand to increase the quality of our sequencing results.

However, we would like to stress the following:

We aim to experimentally dissect the cellular mechanism underlying symbiont-host interaction. To do so, we use various means to form reasonable hypotheses as a starting point for our experimental approaches. In this manuscript, we form a testable hypothesis combing transcriptomic analyses, bioinformatics and integration of previous knowledge from adjacent field such as the better studied field of how pathogens interact with macrophages.

Based on this, we decided to investigate the role of ITA1 in symbiont uptake because:

- (1) Certain integrins and their downstream signalling molecules were differentially regulated upon symbiosis.
- (2) Various human pathogens have co-opted RGD-binding integrin receptors to adhere to, enter, enhance colonization and replication, and spread within the host (42).
- (3) Phylogenetic analysis suggests that ITA1 is the only RGD-binding integrin in Aiptasia.

We therefore hope that the reviewer can appreciate this, and the fact that the cell biology of endosymbiosis is a relatively young field and then focus on the actual phenotypic experiments to functionally dissect integrins in various systems.

Figure 1 and 2: A central point in the paper is that integrins are dynamically regulated during symbiosis formation. Indeed, the discussion presents a model for why dynamic regulation is important for symbiosis control. However, due to the challenges with the RNAseq above, it is hard to tell if that differential regulation is accurate. I recommend orthogonal measurements of integrin expression to support these claims. qPCR of these genes in Apo/Sym larvae and adults and FISH in Apo/Sym larvae would be needed to support the model.

Thanks for this recommendation. We believe that this manuscript provides important evidence that integrin-mediated recognition of a RGD motif plays a role in symbiont uptake, but we would not describe the dynamic regulation of integrin expression as a central point. In contrast, we use our transcriptomic analysis as a hypothesis generator for our detailed experimental work. Indeed, based on our findings we provide ideas for models. However, distinguishing between models is beyond the scope of this paper. Especially, as it is highly unlikely that integrin regulation is purely transcriptional. In the future, it will be interesting to e.g. integrin-specific antibodies to assess protein localization and dynamics in symbiotic cells, ask which down-stream pathways are involved and identify the natural ligands on symbionts. We believe that this paper will spark various analyses in these directions in the community.

Figure 2: It is not clear why ITA1 was the focus. This gene does not appear to be differentially regulated in Figure 1. In situs of the other differentially expressed genes would support the proposed model.

We agree that we did not explain our focus on ITA1 in the original manuscript properly. We have now completely redone this section and added new analyses to improve clarity (for details see above). In brief, our phylogenetic analysis points

directly to ITA1 as this is the only RGD-binding integrin. Accordingly, we have focused on this one for the *in situ* hybridization experiments. We believe that in the context of this paper (aiming to identify a receptor involved in symbionts uptake) this is appropriate while in a different paper (e.g. focusing on a comparative analyses of integrins in Aiptasia or cnidarians) this would be indeed useful.

Figure 2: The chromogenic and fluorescent in situs have different expression patterns. Why is that? Also, higher-resolution/magnification images of each help support determine any differences and the cell types involved. As discussed above, showing differential expression based on infection with mRNA localization is needed in the context of the RNAseq.

We believe the expression patterns for the different methods to be actually quite similar. The representative images may appear somewhat different, because one is a fluorescent signal being detected with a confocal microscope which allows for out of focus light to be not detected, resulting in a clearer image. The chromogenic method is detected using widefield color microscopy, which will still detect out of focus light, resulting in a more blurred image. Due to this difference in Fig 2C you see color coming from the gastroderm layers above and below the gastric cavity, whereas in 2E you only see fluorescence from select slices of the tissue that does not include the inner/outer layers of gastroderm, and here instead you can clearly see the gastric cavity.

Line 173-174: Data not shown - would it be possible to show the data?

We have included these data in the supplement as per EMBO policy.

Line 177-179: A concern with drug and peptide treatment is off-target impacts. Is there orthogonal data that you can provide showing that the peptide should disrupt integrin functions in Aiptasia? An AlphaFold experiment with each of the integrins and each peptide used might help.

We agree that with any pharmacological approach there could be off target effects. However, these peptides have been extensively used to study integrin functions in the literature. Moreover, the bulk of the experiments in the paper validated the involvement of the RGD motif and integrin receptors. Specifically, we use not only RGD-coated beads in Aiptasia but also perform an extensive analysis in human cell culture under highly controlled conditions. This even includes a point-mutation analysis of heterologous integrins that is specific to the RGD-binding mechanism. We are certain that these analyses are far superior than *in silico* modeling approaches and therefore refrained from doing this.

Figure 3-4: Showing only normalized data obscures the experiment variation. Providing raw data and unnormalized plots in the supplement will aid in reproducibility.

We have provided these in the supplement.

Figure 3-4: Statistics - I am not sure that ANOVAs and t-tests are appropriate tests for the normalized data because the data are not normally distributed, violating the test's assumptions. The data in Figure 4D have overlapping SE but are significant, which concerns me. Can the authors clarify?

We have reanalyzed all of our data using R and performed shapiro tests for normality on all the panels in Fig 3 and 4. This revealed that all of our conditions used for ANOVA analysis are normally distributed, except for in figure 3B the DGR 100uM, and from 3K the RGD 100uM and DGR 500uM conditions, the latter two were only modestly non-normally distributed. Therefore, for 3B and 3K we redid the statistics using a test for non-normalized data (permutational analysis of variance) which is similar to ANOVA but it generates the distribution empirically instead of assuming a normal distribution. Here, we get the same result as the ANOVA, where in 3B the significant changes remain and in 3K we find no significance among any of our comparisons. Therefore, we kept the original analysis in these figures. Similarly, all panels where a t-test was performed also had normal distribution via Shapiro test, so the statistics here also remain the same.

Figure 4D: Is there a bead-only control? This control is important to know which peptide is having an impact on symbiosis.

We were unsure of what exactly reviewer means when asking for a "bead-only control". Since stating it would be important to know which peptide impacts symbiosis however, we devised an additional experiment where we coated beads with either DGR or RGD then exposed HEK cells to these beads after transfecting them either with the control CaaX-GFP plasmid or the integrin overexpression plasmids. Here we found that when integrins were overexpressed, RGD-coated beads were phagocytosed 10X more than cells exposed to DGR-coated beads or cells where integrins were not overexpressed. We note that this 10X enhancement in uptake is similar to the 10X increased uptake seen with SSB01 when integrins are overexpressed. These new results are now part of Figure 4D.

Figure 3 F-H: How were algae determined to be inside cells verse in the gut cavity in the coral experiments?

Indeed, coral larvae are relatively thick (approximately 10X the size of Aiptasia larvae) and it is difficult to discern between algae that are inside cells versus floating in the gastric cavity. However, based on previous analysis we have good indications that the algae are inside the host tissue cells. Specifically, we perform multiple washes after a period of co-incubation of larvae and algae to remove surplus algae before fixation and microscopy. In our experience, the algae that remain associated with the coral larvae are indeed firmly integrated rather than floating in the gastric cavity. In addition, we used confocal microscopy to quantify the number of symbionts inside corals which makes it easier to discern between

symbionts inside the tissue and those in the gastric cavity. However, we acknowledge that this still has its limitations and therefore we have updated the manuscript to more cautiously interpret these data by saying algae “inside larvae” instead of “algae uptake” or “phagocytosed algae”.

Figure 4: Is the increase in algal infection specific to YFP-positive cells in A? If not, what is the explanation? Also, how many times was the experiment performed? (Sorry if I missed it).

Unfortunately, non-GFP/YFP cells are difficult to see and count under the microscope, therefore, we only count the fluorescent cells and we believe that the increased uptake is specific to cells that overexpress integrins. We consistently see a baseline level of algae in the field of view that is either in non-fluorescent cells or sticking to the outside of cells. For instance, if you compare Fig. 4A-C there seems to be 4 algae in A, 2 algae in B and 4 in focus algae in C that are not within fluorescent cells. Instead, there is a clear increase in the number of fluorescent cells that house algae in 4A compared to 4B-C.

The experiment shown in 4D has been performed with 3 separate biological replicates, meaning 3 separate weeks worth of different cells and algae. We also have since incorporated this assay into a cell biology practical course we teach here at the LMU to bachelors students, who over the past 3 years of running this course consistently get similar trends in algae uptake across the conditions. Therefore, we believe the assay and the results to be robust and repeatable.

Minor comments:

Introduction:

1. The introduction could be improved by discussing the evidence for and against pre- and post-phagocytic mechanisms while citing the appropriate literature.

We have updated the introduction to discuss pre and post phagocytic mechanisms of symbiont selection

2. Line 115-116: More details about this sentence in the introduction would help the reader frame the study. “Several pathogens rely on receptor-ligand interactions to gain entry into non-phagocytic cells, including several types of integrin interactions (42).”

We have added extra details to the introduction about how pathogens use integrins to invade cells.

Discussion:

Line 298-301: It has been observed that more than one alga can be in a gasterodermal cell. It is not known that only one alga can get into each cell. Can the authors explain this inference?

We believe the line numbers in the version reviewer 3 was provided do not align with our documents, and as such it is difficult for us to determine which statement they are referring to. However, we know from our own studies and those from other labs that individual cells have been found to have multiple symbionts inside, which could be due to taking up multiple algae, OR that the algae once inside a cell divide. Therefore, one of our hypotheses where we state that integrin downregulation may be a mechanism to prevent the cell from taking up multiple algae could be possible, if we assume the cells with multiple algae inside arise from algae proliferation.

Line 369 - Xenia 'soft' coral. I think that will help the evolutionary biologists.

Changed this.

Dear Dr. Guse

Thank you for the submission of your revised manuscript to EMBO reports. I apologize for the delay in handling your manuscript, but we have now received the full set of referee reports that is copied below.

As you will see, all referees are very positive about the study and request only minor changes to clarify text and figures.

From the editorial side, there are also a few things that we need before we can proceed with the official acceptance of your study (see below).

Please provide a point-by-point response to both, the referee and editorial points to ease the final checks.

EDITORIAL POINTS:

- Your manuscript will be published in our Reports section. It seems that you already have a combined Results and Discussion section, but the section title needs to be adjusted accordingly, i.e., to Results and Discussion.
- Please move the Data Availability section before the Acknowledgments.
- You have re-analysed published RNA-seq datasets and could therefore convert (Wolfowicz et al, 2016) and (Jacobovitz et al, 2021) to "Data references". To do so, you would need to cite both, the paper in which the datasets were published and the dataset entry itself.
In the text this would look like: (Wolfowicz et al., 2016, Data ref: Wolfowicz et al, 2016). In the reference list you have the conventional reference to the paper and then in addition - with the same authors - the reference to the Datasets on NCBI or alike, with a URL that resolves directly to the datasets (followed by the tag [DATASET]).
Please see our guide to authors for more information (Data citation, <https://www.embopress.org/page/journal/14693178/authorguide>).
- "Declaration of competing interests" should be renamed to "Disclosure and Competing Interests Statement"
- Regarding the Author Contributions, we now use CRediT to specify the contributions of each author in the journal submission system. Therefore, please remove the Author Contributions from the manuscript file and make sure that the author contributions in our online manuscript tracking system are correct and up-to-date. The information you specified in the system will be automatically retrieved and typeset into the article. You can enter additional information in the free text box provided, if you wish. See also our guide to authors <https://www.embopress.org/page/journal/14693178/authorguide#authorshipguidelines>.
- Funding information: EMBO and Alexander von Humboldt Postdoctoral fellowships should be removed from the Comments box in the online manuscript tracking system and entered as a separate funder; funding in the manuscript should go under Acknowledgments so no separate "Funding" title is needed in the manuscript file .
- Please provide an in-text callout for Figure 3K.
- You have currently 3 EV tables that need to be reformatted as follows:
Table EV2 and EV3 should be removed from the manuscript and uploaded separately as Table EV1 and Table EV2.
The current Table EV1 is complex and constitutes a dataset. Therefore, please call it Dataset EV1 and upload it as file type Dataset. The legends of this dataset provided in a Word file need to be in the same Excel file, provided as a separate sheet/tab.
- Please remove the Reagents and Tools table from the manuscript and upload it as a separate Word file (type Reagent table).
- Text from lines 21-23 needs to be removed from the title page of the manuscript file.
- Methods should be used as the section heading for that section.
- Please address the following comments in the figure legends:
 - > note the exact p values in the legend of figure EV4.
 - > define the box plots in terms of minima, maxima, centre and percentile in the legend of figure EV4.
 - > add information related to n in the legend of figure 3H.
- What about a more 'direct' title such as:
"Integrins mediate symbiont-specific uptake in cnidarian larvae"

- I have introduced a few minor edits in the Abstract, mainly to describe findings in present tense. Please see my suggested Abstract below my signature.

With kind regards,

=====

Referee #1:

The authors have addressed my comments satisfactorily and I am happy to support publication of the current version.

Referee #2:

The revised manuscript "Symbiont-specific uptake is mediated by integrins in cnidarian larvae" from Jones et al. is much improved over the first version; the authors addressed all major points and most minor comments and greatly enhanced readability of the manuscript. There are still some minor points that need to be addressed but overall, the manuscript should be suitable for publication after doing so and should be of interest to a broad audience.

Minor comments:

1. The usage of gene/transcript vs. protein names is still inconsistent and incorrect, especially the paragraphs in lines 128-149, 174-199, and 578-607, where gene names (italicized) are used but the remainder of the sentence talks about proteins or vice versa.
2. Figure 1 really benefited from the re-analysis and manual curation, but the legend should contain a short explanation for the empty cells in the differential expression (now only in the Material & Methods); that would prevent any confusion.
3. A control without any treatment would be good to show as reference for the *A. digitifera* larvae as was done for the ones from *Aiptasia* or at least some contextualization with previous experiments/studies; the difference between DGR and RGD-treated larvae is pretty minimal, and it makes a difference if the normal uptake is in the hundreds of algae or close/identical to the DGR peptide numbers.
4. I would suggest swapping panels B and C in Fig. 4 to align with the order they are mentioned in the text.
5. The y-axis in Fig. EV2H should start at 0 to not overemphasize the actual differences.

Line-specific comments:

L145: ", multiple genes that encode NPC2,..."
- Should be protein name NPC2.

L159: "symbiont containing"
- symbiont-containing

L165: "cell intrinsic"
- cell-intrinsic

L168-169: "...to shift the functionality of the cell from adhesion to symbiosis establishment."
- I would recommend rephrasing this sentence; currently the first thing coming to mind is that the gastrodermal cells take up symbionts and detach from the gastroderm.

L180: "as well as a single gene from each cnidarian species"
- All access numbers and descriptions in the previous and following sentences refer to proteins.

L195-196: "Therefore, we determined the location of the predicted RGD-binding integrin, ITA1, expression using in situ

hybridization (ISH) and fluorescence in situ hybridization (FISH) on Aiptasia larvae."

- The RGD-binding integrin refers to the protein (non-italicized).

- The sentence structure could be improved: "Therefore, we localized the transcripts of the predicted RGD-binding integrin ITA1 by chromogenic in situ hybridization (ISH) and fluorescence in situ hybridization (FISH)."

L304-306: "RGD-coated beads were taken up by integrin overexpressing cells approximately ten times more efficiently than DGR-coated beads or cells where integrins were not overexpressed (Fig. EV4)."

- "RGD-coated beads were taken up by integrin-overexpressing cells approximately ten times more efficiently than DGR-coated beads or by cells in which integrins were not overexpressed (Fig. EV4)."

L317: "Our data suggests integrin-ligand..."

- "Our data suggest integrin-ligand..."

L324: "...and is an area of potential future research."

- Better: "...which is an area of potential future research."

Reagents and Tools Table: "Diago's IMK medium"

- "Daigo's IMK medium"

L530: "Sp6 polymerase"

- Still needs to be "SP6 polymerase"

L631: "...aligned on..."

- "...aligned to..."

Referee #3:

The authors did a reasonable job with the proposed suggestions.

=====

Abstract

Endosymbiosis between dinoflagellate algae and cnidaria is fundamental for coral reef health. Appropriate symbiont selection is required for sufficient host nutrient acquisition and could be tailored to increase cnidarian stress tolerance. Previous research suggested glycan-lectin interactions facilitate symbiont uptake; however, blockage of such interactions does not fully inhibit symbiosis establishment, suggesting other receptors are at play. Here, we use a combination of cnidarian model systems and human cell lines to determine if phagocytic integrins facilitate symbiont recognition and uptake. Integrins are highly expressed in the gastrodermal tissue of the host, where symbiosis takes place, and symbiont uptake alters the expression of integrins and downstream signalling molecules. Blockage of integrin binding sites with competitor peptides reduces symbiont uptake, while uptake of non-symbiotic algae, or uptake in a non-symbiotic cnidarian is unaffected. Finally, overexpression of phagocytic integrins in human cells increases symbiont uptake and mutation of the active binding site abolishes uptake. Our findings reveal integrins as important receptors for symbiosis establishment and shed light on the evolutionary functions of integrins during phagocytosis.

Point by point response

EDITORIAL POINTS:

- Your manuscript will be published in our Reports section. It seems that you already have a combined Results and Discussion section, but the section title needs to be adjusted accordingly, i.e., to Results and Discussion. We have made this change

- Please move the Data Availability section before the Acknowledgments. We have made this change

- You have re-analysed published RNA-seq datasets and could therefore convert (Wolfowicz et al, 2016) and (Jacobovitz et al, 2021) to "Data references". To do so, you would need to cite both, the paper in which the datasets were published and the dataset entry itself. In the text this would look like: (Wolfowicz et al., 2016, Data ref: Wolfowicz et al, 2016). In the reference list you have the conventional reference to the paper and then in addition - with the same authors - the reference to the Datasets on NCBI or alike, with a URL that resolves directly to the datasets (followed by the tag [DATASET]). Please see our guide to authors for more information (Data citation, <https://www.embopress.org/page/journal/14693178/authorguide>). We have updated this

- "Declaration of competing interests" should be renamed to "Disclosure and Competing Interests Statement" We have made this change

- Regarding the Author Contributions, we now use CRediT to specify the contributions of each author in the journal submission system. Therefore, please remove the Author Contributions from the manuscript file and make sure that the author contributions in our online manuscript tracking system are correct and up-to-date. The information you specified in the system will be automatically retrieved and typeset into the article. You can enter additional information in the free text box provided, if you wish. See also our guide to authors <https://www.embopress.org/page/journal/14693178/authorguide#authorshipguidelines>. We have removed this section from the text and verified the correctness of the submitted data.

- Funding information: EMBO and Alexander von Humboldt Postdoctoral fellowships should be removed from the Comments box in the online manuscript tracking system and entered as a separate funder; funding in the manuscript should go under Acknowledgments so no separate "Funding" title is needed in the manuscript file . We have updated the funding section in the online form and moved the funding info in the text file to the acknowledgements section.

- Please provide an in-text callout for Figure 3K. This callout is on line 236

- You have currently 3 EV tables that need to be reformatted as follows: Table EV2 and EV3 should be removed from the manuscript and uploaded separately as Table EV1 and Table EV2. The current Table EV1 is complex and constitutes a dataset. Therefore, please call it Dataset EV1 and upload it as file type Dataset. The legends of this dataset provided in a Word file need to be in the same Excel file, provided as a separate sheet/tab.

- Table EV2 and EV3 have been removed and uploaded as separate Word files. Dataset EV1 has been uploaded with the legend as a separate tab in the Excel file.

- Please remove the Reagents and Tools table from the manuscript and upload it as a separate Word file (type Reagent table). We have done this

- Text from lines 21-23 needs to be removed from the title page of the manuscript file. We removed these lines.

- Methods should be used as the section heading for that section. We updated this

- Please address the following comments in the figure legends:

-> note the exact p values in the legend of figure EV4.

-> define the box plots in terms of minima, maxima, centre and percentile in the legend of figure EV4.

-> add information related to n in the legend of figure 3H.

The n has been added to the figure legend for 3H. The boxplot definition has been updated for Fig. EV4 and the exact p values have been added to the legend.

- **What about a more 'direct' title such as:**

"Integrins mediate symbiont-specific uptake in cnidarian larvae"

- **I have introduced a few minor edits in the Abstract, mainly to describe findings in present tense. Please see my suggested Abstract below my signature.**

Thank you for the suggestions, we have made these changes.

With kind regards,

=====

Referee #1:

The authors have addressed my comments satisfactorily and I am happy to support publication of the current version.

Referee #2:

The revised manuscript "Symbiont-specific uptake is mediated by integrins in cnidarian larvae" from Jones et al. is much improved over the first version; the authors addressed all major points and most minor comments and greatly enhanced readability of the manuscript. There are still some minor points that need to be addressed but overall, the manuscript should be suitable for publication after doing so and should be of interest to a broad audience.

Minor comments:

1. The usage of gene/transcript vs. protein names is still inconsistent and incorrect, especially the paragraphs in lines 128-149, 174-199, and 578-607, where gene names (italicized) are used but the remainder of the sentence talks about proteins or vice versa.

We have updated these sections to more accurately refer to protein/RNA.

2. Figure 1 really benefited from the re-analysis and manual curation, but the legend should contain a short explanation for the empty cells in the differential expression (now only in the Material & Methods); that would prevent any confusion.

We have added an explanation in the legend for these empty cells.

3. A control without any treatment would be good to show as reference for the *A. digitifera* larvae as was done for the ones from *Aiptasia* or at least some contextualization with previous experiments/studies; the difference between DGR and RGD-treated larvae is pretty minimal, and it makes a difference if the normal uptake is in the hundreds of algae or close/identical to the DGR peptide numbers.

Unfortunately, we do not have the data for larvae without any peptide added as a control. Comparison with our previous *Acropora* larvae data is difficult as well, since in this paper we performed short exposures of 24hr and counted symbionts per larvae, whereas in previous studies incubation times were typically longer (10 days in Wolfowicz et al. 2016) and there the measurement was percent of infected larvae. Therefore, we have not added to the text to compare across these different datasets. Regardless, the DGR peptide is the proper control to be comparing to and we believe our analysis still holds true.

4. I would suggest swapping panels B and C in Fig. 4 to align with the order they are mentioned in the text.

We have swapped these panels and updated the text accordingly.

5. The y-axis in Fig. EV2H should start at 0 to not overemphasize the actual differences.

We have made this change.

Line-specific comments:

L145: ", multiple genes that encode NPC2,..."

- Should be protein name NPC2.

We have made this change.

L159: "symbiont containing"

- **symbiont-containing**

We have made this change.

L165: "cell intrinsic"

- **cell-intrinsic**

We have made this change.

L168-169: "...to shift the functionality of the cell from adhesion to symbiosis establishment."

- I would recommend rephrasing this sentence; currently the first thing coming to mind is that the gastrodermal cells take up symbionts and detach from the gastroderm.

We updated it to now say "while those involved in phagocytosis maintain their expression to prioritize symbiosis establishment."

L180: "as well as a single gene from each cnidarian species"

- All access numbers and descriptions in the previous and following sentences refer to proteins.

We have changed this to instead refer to proteins.

L195-196: "Therefore, we determined the location of the predicted RGD-binding integrin, ITA₁, expression using in situ hybridization (ISH) and fluorescence in situ hybridization (FISH) on Aiptasia larvae."

- The RGD-binding integrin refers to the protein (non-italicized).

- The sentence structure could be improved: "Therefore, we localized the transcripts of the predicted RGD-binding integrin ITA₁ by chromogenic in situ hybridization (ISH) and fluorescence in situ hybridization (FISH)."

We have made this change.

L304-306: "RGD-coated beads were taken up by integrin overexpressing cells approximately ten times more efficiently than DGR-coated beads or cells where integrins were not overexpressed (Fig. EV4)."

- "RGD-coated beads were taken up by integrin-overexpressing cells approximately ten times more efficiently than DGR-coated beads or by cells in which integrins were not overexpressed (Fig. EV4)."

We have made this change.

L317: "Our data suggests integrin-ligand..."

- "Our data suggest integrin-ligand..."

We have made this change.

L324: "...and is an area of potential future research."

- Better: "...which is an area of potential future research."

We have made this change.

Reagents and Tools Table: "Diago's IMK medium"

- "Daigo's IMK medium"

We have made this change.

L530: "Sp6 polymerase"

- Still needs to be "SP6 polymerase"

We have made this change.

L631: "...aligned on..."

- "...aligned to..."

We have made this change.

Referee #3:

The authors did a reasonable job with the proposed suggestions.

=====

Abstract

Endosymbiosis between dinoflagellate algae and cnidaria is fundamental for coral reef health. Appropriate symbiont selection is required for sufficient host nutrient acquisition and could be tailored to increase cnidarian stress tolerance. Previous research suggested glycan-lectin interactions facilitate symbiont uptake; however, blockage of such interactions does not fully inhibit symbiosis establishment, suggesting other receptors are at play. Here, we use a combination of cnidarian model systems and human cell lines to determine if phagocytic integrins facilitate symbiont recognition and uptake. Integrins are highly expressed in the gastrodermal tissue of the host, where symbiosis takes place, and symbiont uptake alters the expression of integrins and downstream signalling molecules. Blockage of integrin binding sites with competitor peptides reduces symbiont uptake, while uptake of non-symbiotic algae, or uptake in a non-symbiotic cnidarian is unaffected. Finally, overexpression of phagocytic integrins in human cells increases symbiont uptake and mutation of the active binding site abolishes uptake. Our findings reveal integrins as important receptors for symbiosis establishment and shed light on the evolutionary functions of integrins during phagocytosis.

Prof. Annika Guse
Ludwig-Maximilian-Universität
Germany

Dear Prof. Guse,

I am very pleased to accept your manuscript for publication in the next available issue of EMBO reports. Thank you for your contribution to our journal.

Yours sincerely,
